# Uncoupling the roles of firing rates and spike bursts in shaping the STN-GPe beta band oscillations

**Jyotika Bahuguna**[1☯]*, **Ajith Sahasranamam**[2☯], **Arvind Kumar**[3]*

**1** Aix Marseille University, Institute for Systems Neuroscience, Marseille, France, **2** Ongil Pvt Ltd, Singapore, **3** Department of Computational Science and Technology, School of Electrical Engineering and Computer Science, KTH Royal Institute of Technology, Stockholm, Sweden

☯ These authors contributed equally to this work.
* jyotika.bahuguna@gmail.com (JB); arvkumar@kth.se (AK)

**Data Availability Statement:** This project is shared on open science framework (OSF - https://osf.io/quycb/). The project consists of the figures not included in the manuscript namely: a) An example of spectral entropy mentioned in section Spectral

## Abstract

The excess of 15-30 Hz ($\beta$-band) oscillations in the basal ganglia is one of the key signatures of Parkinson's disease (PD). The STN-GPe network is integral to generation and modulation of $\beta$ band oscillations in basal ganglia. However, the role of changes in the firing rates and spike bursting of STN and GPe neurons in shaping these oscillations has remained unclear. In order to uncouple their effects, we studied the dynamics of STN-GPe network using numerical simulations. In particular, we used a neuron model, in which firing rates and spike bursting can be independently controlled. Using this model, we found that while STN firing rate is predictive of oscillations, GPe firing rate is not. The effect of spike bursting in STN and GPe neurons was state-dependent. That is, only when the network was operating in a state close to the border of oscillatory and non-oscillatory regimes, spike bursting had a qualitative effect on the $\beta$ band oscillations. In these network states, an increase in GPe bursting enhanced the oscillations whereas an equivalent proportion of spike bursting in STN suppressed the oscillations. These results provide new insights into the mechanisms underlying the transient $\beta$ bursts and how duration and power of $\beta$ band oscillations may be controlled by an interplay of GPe and STN firing rates and spike bursts.

## Author summary

The STN-GPe network undergoes a change in firing rates as well as increased bursting during excessive $\beta$ band oscillations during Parkinson's disease. In this work we uncouple their effects by using a novel neuron model and show that presence of oscillations is contingent on the increase in STN firing rates, however the effect of spike bursting on oscillations depends on the network state. In a network state on the border of oscillatory and non-oscillatory regime, GPe spike bursting strengthens oscillations. The effect of spike bursting in the STN depends on the proportion of GPe neurons bursting. These results suggest a mechanism underlying a transient $\beta$ band oscillation bursts often seen in experimental data.

Entropy) b) Individual trials of the spectrograms of the average spectrogram (more details in section Control of the amplitude and duration of β band oscillation bursts by spike bursting). The project also consists of a git link (https://github.com/jyotikab/stn_gpe_ssbn) to the code required to simulate and analyze the model.

**Funding:** This work was funded by Swedish Research Council (StratNeuro and India-Sweden collaboration grant and VR research project grant), KTH digital futures: dBRAIN and German Research Foundation (DFG; grant DI 1721/3-1 [KFO219-TP9]). The funders had no role in study design, data collection and analysis, decision to publish, or preparation of the manuscript.

**Competing interests:** I have read the journal's policy and the authors of this manuscript have the following competing interests: We would like to inform you that one of the authors, Ajith Sahasranamam is affiliated with Ongil Pvt Ltd, however for the work under consideration for publication, that there is no conflict of interest with the business activities of the company - Ongil Pvt Ltd. The Ongil Pvt Ltd also played no role in designing and financing this work.

# Introduction

Parkinson's disease (PD) is a progressive neurodegenerative brain disease caused by the depletion of dopamine neurons in the substantia nigra pars compacta (SNc) [1]. Loss of dopamine causes a host of cognitive and motor impairments. Dopaminergic cell death can be attributed to many causes e.g. genetic mutations [1], pathogen that affects the gut microbome and travels to the central nervous systems [2, 3], excitotoxicity [4], and mitochondrial dysfunction [5] etc. [6]. While the etiology of PD is still debated, the behavioral symptoms of PD are accompanied by various changes in the neuronal activity in Basal Ganglia (BG): e.g, increased firing rate of D2 type dopamine receptors expressing striatal neurons [7–9]; increase in spike bursts in striatum, globus pallidus externa (GPe), globus pallidus interna (GPi) and subthalamic nuclei (STN) [8] and increased synchrony in all BG nuclei [10] including striatum [11], GPe [12, 13], STN [14–16] and GPi/SNr [12, 17, 18]. Besides these changes in neuronal activity, at the population level, there is an increase in the power and duration of β band oscillations (15-30 Hz) in local field potential (LFP) recorded from the basal ganglia of PD patients [14, 18–20]. The β band oscillations are mainly correlated with motor deficits such as rigidity, bradykinesia and akinesia [14, 16, 21] and, suppression of these oscillations, for example, by deep brain stimulation (DBS) ameliorates motor symptoms of PD. Therefore, there is a great interest in understanding the mechanisms underlying the origin of β band oscillations which are not well understood. For instance, it is unclear whether the oscillations are imposed by cortical inputs [22–24] or they are generated within the BG, either in striatum [25], in pallidostriatal circuit [26] or the STN-GPe network [8, 27–35]. Several experimental results indicate that GPe-STN network plays an integral role in generating and modulating these oscillations [14, 18, 19, 36] and their stimulation have been shown to affect (disrupt/modulate) oscillations [8, 37, 38].

From a dynamical systems perspective, interaction between excitatory and inhibitory neuronal population form the necessary substrate for oscillations where an imbalance of timing and/or strength of effective excitation and inhibition leads to population oscillations [39, 40]. Several excitatory and inhibitory loops can be identified in the BG which may underlie the emergence of β band oscillations among which STN-GPe circuit has emerged as a primary candidate. In both firing rate-based and spiking neuronal network models, an increase in the coupling between STN and GPe is sufficient to induce strong oscillations [28, 31, 33]. However, the oscillations may also be created if effective excitatory input to STN neurons (from the cortex) or effective inhibitory input to GPe neurons (from the striatum) is increased [29, 35]. Besides, the GPe-STN network, the imbalance of the direct (effectively excitatory) and hyperdirect (effectively inhibitory) pathways of the BG can also cause oscillations [41]. These computational models not only suggest possible mechanisms underlying the β oscillations but also provide explanations for the altered synaptic connectivity within the BG and how increased firing rates in the striatal neuron projecting to the GPe [7] can lead to pathological oscillations.

Recent data from human patients suggest that β band oscillations are not persistent and occur in short epochs which are called β oscillation bursts [20]. The β oscillation bursts in fact, might appear as persistent oscillations as a result of averaging over multiple trials [42–44] in order to account for the inter-trial variability. Such β oscillation bursts have also been observed in healthy animals in certain task conditions [45]. In human patients, characteristics of β-oscillation bursts are associated with motor performance [20, 46, 47]. The presence of oscillatory bursts might be a result of transient change in external input [35] or phase slips between the population activity of STN and GPe [48]. In general, however, the mechanisms due to which these oscillatory bursts arise are unclear.

The $\beta$ band oscillations are also accompanied by an increase in spike bursting along with the firing rate changes. In MPTP models of non-human primates, the proportion of bursty spikes in STN and GPe is significantly higher in animals with PD than the healthy animals [8, 49]. Increased spike bursting in GPe and STN is also observed in 6-OHDA treated rodents [50, 51]. But it remains unclear how increased spike bursting affects the duration and power of $\beta$ band oscillations.

However, it should be noted neurons in the STN-GPe network show spike bursting even in healthy conditions [8, 49]. Therefore, it is important to understand whether the spike bursting and the pathological oscillations share a causal relationship and if this is the case, then why spike bursts are also observed in healthy states [8, 49]. In addition, it is also crucial to tease apart the contribution of altered firing rates and spike bursting to the $\beta$-band oscillations to better understand the pathophysiology of PD and find better way to quench the pathological oscillations.

To understand the role of spike bursting in shaping the beta oscillations here, we investigated the effect of firing rates and patterns on the presence of oscillations using a computational model of the STN-GPe network. Usually, the average firing rate of a neuron is tightly coupled to spike bursting and it is not easy to disentangle the effect of these two variables independently. To solve this we used the State-dependent Stochastic Bursting Neuron Model (SSBN) model [52], which allowed us to vary firing rate and firing pattern (spike bursting) of the neuron independently and hence uncouple the effects of firing rate and spike bursting on the $\beta$ band oscillations.

Using the model, we found that the average firing rate of STN neurons was predictive of oscillations but surprisingly, the average firing rate of GPe neurons was not. Notably, the changes in firing rate of STN and GPe neurons resulted in persistent oscillations in the $\beta$ band. The effect of GPe and STN spike bursting on STN-GPe oscillations was however, state dependent. When the network exhibited strong oscillations or aperiodic activity, spike bursting in STN and GPe had no effect on the global state of network activity. However, in the regime at the border of oscillatory and non-oscillatory states (transition regime), an increase in the fraction of bursting neurons in GPe, enhanced oscillations. By contrast, small to moderate fraction of bursting neurons in STN quenched the oscillations whereas when most of the STN neurons were bursting, network re-exhibited strong oscillations. Furthermore, in the transition regime, when a small fraction of GPe and STN neurons were bursty, $\beta$ band oscillations occurred in short epochs that closely resembled with the population activity as observed in the experimental data. Thus, our model suggests that spike bursting may be one of the mechanisms to generate these $\beta$-oscillation bursts ($\beta$-bursts). Taken together, these results for the first time, separate the roles of firing rates and spike bursting and shows how spike bursting in the STN and GPe can either enhance or suppress the $\beta$ band oscillations, depending on the network activity state. That is, the nature of $\beta$ band oscillations is jointly determined by a combination of the underlying network state and proportion of neurons that are bursty. Finally, our results revealed that STN and GPe may play a qualitatively different roles in shaping the dynamics of beta band oscillations. These insights suggest new means to quench the pathological oscillations.

## Materials and methods

### Neuron model

In the existing reduced neuron models (e.g. leaky-integrate-fire neuron), to achieve changes in the firing patterns, the sub-threshold dynamics of the neuron model needs to be altered. However, when a neuron model is modified to exhibit spike bursting, its input-output firing rate

relationship ($f - I$ curve) is also altered. That is, spike bursting and neuron firing rate are coupled and prevent the comparison with non-bursting neuron with the same firing rate. However, to isolate the effect of changes in the firing patterns on the network dynamics, the $f - I$ curve of the neuron and its firing pattern need to be independently controlled. To achieve this, we use the State-dependent Stochastic Bursting Neuron (SSBN) [52]. The subthreshold membrane potential dynamics of the SSBN model is same as that of the Leaky Integrate and Fire (LIF) neuron:

$$\tau_m \dot{V}_m = -V_m + I_{syn}$$

where, $\tau_m$ is the membrane time constant, $V_m$ is the membrane potential and $I_{syn}$ is the total synaptic current to the neuron. The spike generation mechanism of SSBN is stochastic. On reaching the spiking threshold $V_{th}$, the SSBN generates a burst of $b$ spikes with a probability of $1/b$ every time $V_m \geq V_{th}$. This allows us to vary the size of spike burst without affecting the spike rate and the input output neuron transfer function of the neuron (in S1 Fig). The inter-spike-interval within the burst is constant and is same as the refractory period of the neuron ($5ms$). In order to ensure that the qualitative results are independent of the choice of the refractory period, we reproduced one of the figures for two additional values of refractory periods, $3ms$ and $7ms$. The details are discussed in the section *State dependent effect of spike bursting neurons on β band oscillations*.

More details about this neuron model can be found in [52]. All the neurons in the STN and GPe were modelled as SSBNs. The neuron parameters used are consistent with the STN-GPe network used in a recent work by [35] and are listed in Table 1. We used the same neuron parameters for STN and GPe neurons, however the two neuron types received different amount of external inputs as we explored network state space for different external inputs to the GPe and STN.

## Synapse model

Synapses were modelled as a transient change in conductance. Each spike elicited an alpha-function shaped change in the post-synaptic conductance. The reversal potential determined whether the synapse was excitatory or inhibitory. The peak conductance of each type of synapse is provided in the Fig 1 and Table 2 and the excitatory and inhibitory time constants are

**Table 1. Neuron parameters as used in [35].**

| Parameter | Value | Description |
|:---:|:---:|:---:|
| $C_m$ | $200pF$ | Membrane capacitance |
| $\tau_m$ | $20ms$ | Membrane Time Constant |
| $V_{th}$ | $-54mV$ | Firing threshold |
| $V_{reset}$ | $-70mV$ | Reset potential |
| $\tau_{ref}$ | $5ms$ | Refractory period |
| $B_{isi}$ | $5ms$ | Inter-spike interval within a spike burst |
| $B$ | 1 or 4 | Number of spikes in a burst |
| $\tau_{exc}$ | $5ms$ | Excitatory synaptic time constant |
| $\tau_{inh}$ | $10ms$ | Inhibitory synaptic time constant |
| $g_L$ | $10nS$ | Leak conductance |
| $E_{ex}$ | $0mV$ | Reversal potential (excitatory) |
| $E_{in}$ | $-80.0mV$ | Reversal potential (inhibitory) |

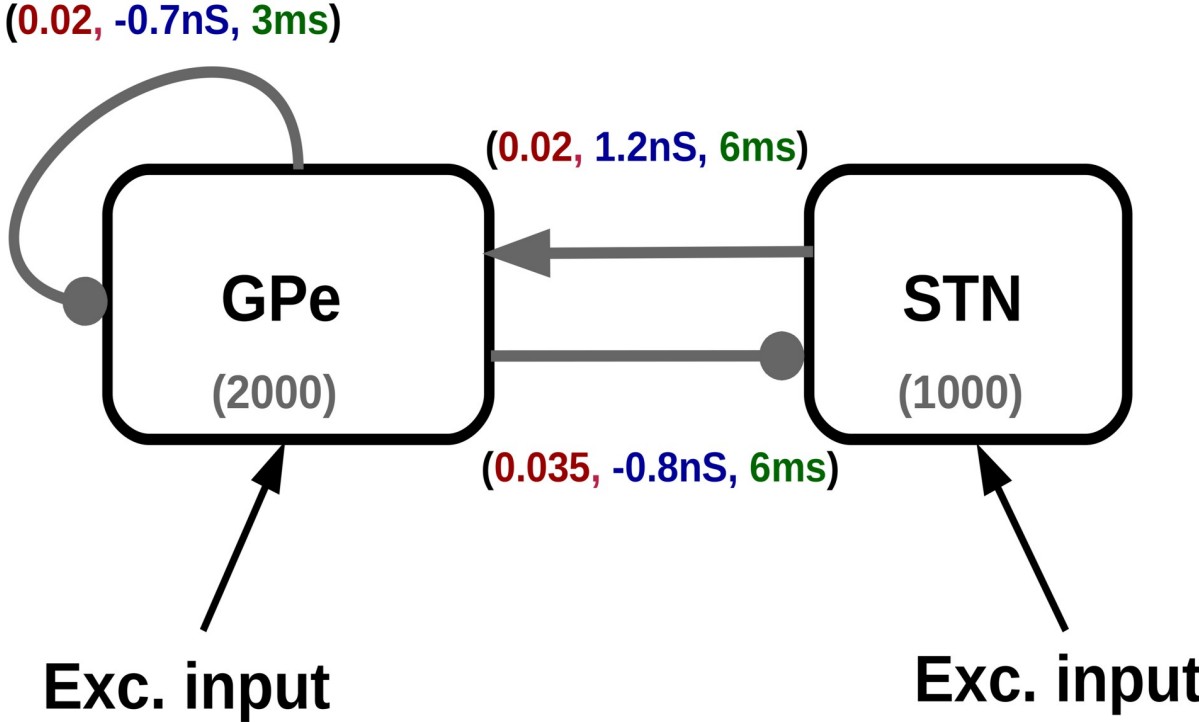

**Fig 1. Schematic of the STN-GPe network.** The connection probability, synaptic strength and delay for each connection is shown in red, blue and green, respectively. The number in parentheses (1000, 2000) represent the number of neurons in STN and GPe, respectively. The connection with arrowhead are excitatory and those with filled circle are inhibitory. The F-I curves for the neuron model with different spike burst lengths is plotted in S1 Fig. The inter spike interval within the burst is kept constant.

shown in Table 1. For further details on dynamics, refer to "iaf_cond_alpha" neuron model in NEST [53].

## STN-GPe network model

The network model consisted of 2000 inhibitory (corresponding to the GPe population) and 1000 excitatory (corresponding to the STN population) neurons. The neurons were connected in a random manner with fixed connection probabilities. The connection strength, connection

**Table 2. Network parameters as used in [35]** The median, 25% and 75% quartiles of the distributions are reported in brackets.

| Parameter | Value | Description |
| --- | --- | --- |
| $\epsilon_{gpe-gpe}$ | 0.02 (0.018,0.015, 0.021) | GPe to GPe connectivity |
| $\epsilon_{gpe-stn}$ | 0.035 (0.036,0.032, 0.04) | GPe to STN connectivity |
| $\epsilon_{stn-gpe}$ | 0.02 (0.02,0.017, 0.023) | STN to GPe connectivity |
| $J_{gpe-gpe}$ | $-0.7nS$ $(-0.68, -0.81, -0.57)$ | GPe to GPe synaptic strength |
| $J_{gpe-stn}$ | $-0.8nS$ $(-0.83, -0.93, -0.72)$ | GPe to STN synaptic strength |
| $J_{stn-gpe}$ | $1.2nS$ (1.0,0.85, 1.2) | STN to GPe synaptic strength |
| $\tau_{gpe-gpe}$ | $3.0ms$ (2.8 2.5, 3.2) | GPe to GPe synaptic delay |
| $\tau_{gpe-stn}$ | $6.0ms$ (5.7, 4.7, 6.4) | GPe to STN synaptic delay |
| $\tau_{stn-gpe}$ | $6.0ms$ (5.7, 4.9, 6.4) | STN to GPe synaptic delay |

probability and synaptic delays were identical to the one used in the model by Mirzaei et al. [35] and are shown in Fig 1.

We investigated the oscillation dynamics of the STN-GPe network in two conditions:

**Condition A**: To characterize the effect of firing rates on $\beta$ band oscillations we studied the network when all the neurons were non-bursting type. For these simulations we set $B = 1$ for all the neurons.

**Condition B**: To characterize the effect of spike bursting on $\beta$ band oscillations we used networks in which a fraction of STN and/or GPe neurons were bursting type. The fraction of bursting neurons in the two populations was varied systematically from 0 to 1. For these simulation we set the spike burst length $B = 4$ for the bursting neurons and $B = 1$ for the non-bursting (or regular spiking neurons).

**Robustness analysis of network parameters.**   In order to ensure that our results are not dependent on a specific choice of network parameters used in [35], we also performed a robustness analysis. To this end we simulated 10000 different models. For each model the value of each of the model parameters (i.e. network connection probability, synaptic strength and delays) were drawn from a Gaussian distribution, whose mean was set to the value used in the model by Mirzaei et al. [35] and the standard deviation was taken as 20% of the mean value. For each parameter set (comprising of nine model parameters -see Table 2), the model was simulated for different values of external input to STN and GPe neurons to generate different network activity states, characterized by their value of spectral entropy. The range of STN and GPe inputs was same as used to generate the results shown in Fig 2. Each model was simulated five times with different random number seeds. The spectral entropy for the five trials was averaged to obtain the state space (e.g. Fig 2C) for each parameter set. Next, we identified the parameter set that results in a state space which had linearly separable oscillatory (spectral entropy $\leq 0.45$) and non-oscillatory (spectral entropy $\geq 0.55$) regions. This was done using a Support Vector Classifier (SVC) from python library *sklearn* to classify our networks into two classes (class label 0: non-oscillatory, spectral entropy $\geq 0.55$ and class label 1: oscillatory, spectral entropy $\leq 0.45$). Using this analysis we retained the models that resulted in a classification score of 1. From the retained models, we estimated the distribution of each network parameter.

We would however like to point out, that this is a preliminary robustness analysis and is no way a comprehensive sensitivity analysis which may include calculation of sensitivity of different features (e.g. spectral entropy) with respect to perturbations in network parameters, analysis of the "sloppy"/sensitive parameters and/or covariance between the parameter values [54, 55], which is beyond the scope of the this work.

## Input

All neurons in the STN and GPe received external excitatory input which was modelled as uncorrelated Poisson spike trains. This input was tuned to match the range of firing rates of the STN and GPe observed in *in vivo* data during healthy and Parkinsonian conditions [19, 35, 51].

To characterize the role of firing rates simulations (condition A) we systematically varied the rate of Poisson spike trains independently for the STN and GPe neurons. For each parameter set we performed at least 5 trials with different random seeds.

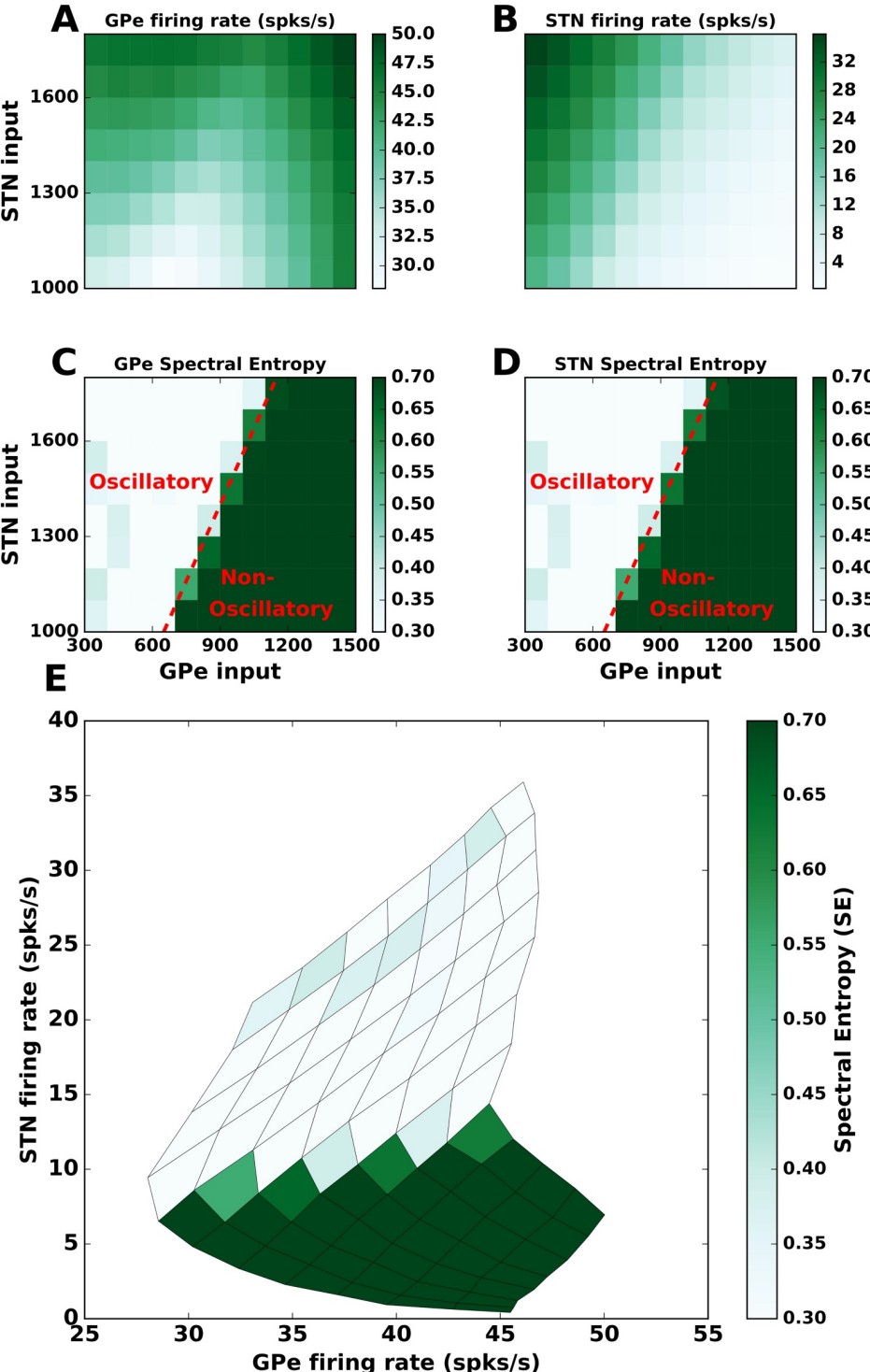

**Fig 2. Effect of STN and GPe firing rates on _β_ band oscillations. (A)** Average firing rate of GPe neurons as a function of different input rates to the STN and GPe. **(B)** Same as in **A** but for STN neurons. **(C)** Strength of oscillations in the GPe population (quantified using spectral entropy, see Methods). **(B)** Same as in **C** but for STN neurons. **(E)** The effect of the STN and GPe firing rates (as in **A** and **B**) on spectral entropy (as in **C** and **D**). These results show that _β_ band oscillations in the STN-GPe network depend on the STN firing rate but not on the GPe firing rates. All the values (firing rate and spectral entropy) were averaged over 5 trials. A scatter plot for spectral entropy against the STN and GPe firing rates for all the 5 trials is shown in S4 Fig.

### Data analysis

**Spectrum of the population activity.**    To estimate the spectrum of the network activity we binned (bin width = 5 ms) the spiking activity of all the STN or GPe neurons to obtain the population activity ($S$). We subtracted the mean and estimated the spectrum ($P$) using the fast Fourier transform (frequency resolution = 5 Hz). To estimate spectral entropy (see below) we measured the $P$ for the whole duration of simulations (7500 ms). To estimate the time-resolved spectrum we measured $P$ for sliding windows (window size = 200ms; overlap = 50ms).

**Spectral entropy.**    To quantify how oscillatory the network activity was, we computed the spectral entropy $H_S$, which is a measure of dispersion of spectral energy of a signal [52, 56].

$$H_S = \frac{-\sum_k P_k \log P_k}{\log N}$$

where $P_k$ is the spectral power at frequency $k$ and $N$ is the total number of frequency bins considered. To estimate spectral entropy we normalized $P_k$ such that $\Sigma_k P_k = 1$. The spectral power was calculated in the $\beta$ frequency range, i.e., 10-35 Hz [57]. Note that we consider the frequency range wide enough to cover both low ($10 - 20 Hz$) and high beta ($20 - 35 Hz$) as we span across a wide range of input firing rates. There is a large variability in the peak frequency and range of the $\beta$-band oscillations even among rodents. It may depend on the pathological state of the animal (healthy or 6-OHDA lesioned) and/or the recording conditions (anaesthetized or awake). In healthy rats, the GPe-LFPs recorded in rats during quiet rest peaked around $13 - 17 Hz$ across animals [58]. In 6-OHDA lesioned anaesthetized rats, mean peak frequency for GPe-LFPs is reported to be $\approx 17 - 22 Hz$ [13, 59] and $16 - 21 Hz$ for STN-LFPs [59]. In awake behaving rats beta band oscillations are faster e.g. STN-LFPs mean beta band peak frequency lies between $22 - 36 Hz$ [15, 60]. To cover all these cases, we set a broad range (10 $- 35 Hz$) to calculate the spectral power [57].

An aperiodic signal (e.g. white noise) for which the spectral power is uniformly distributed over the whole frequency range, has $H_S = 1$. By contrast, periodic signals that exhibit a peak in their spectrum (e.g. in the $\beta$ band) have lower values of $H_S$. In an extreme case, for a single frequency sinusoidal signal $H_S = 0$. Thus, $H_S$ varies between 0 and 1. A simple demonstration of the measure of spectral entropy for the effect of noise and multiple peaks is available at the following weblink (https://osf.io/quycb/—Figures/Spectral_entropy_example). This includes the figure and the corresponding script required to reproduce the figure.

**Duration and amplitude of beta oscillations bursts (beta bursts).**    We defined the length of a burst of $\beta$ band oscillations ($\beta$-burst) as the duration for which instantaneous power in the $\beta$ band remained above a threshold ($\beta_{th}$). $\beta_{th}$ was the average power in the $\beta$ band for an uncorrelated ensemble of Poisson type spikes trains with same average firing rate as the neurons in the network. Because neurons in our model had different average firing rate (averaged over 5 trials) depending on the external input and network activity states, $\beta_{th}$ for each network activity state was different. The $\beta$ oscillation burst amplitude was estimated as the peak power in the $\beta$ band. To estimate the $\beta$ oscillation burst amplitude we smoothened the power spectrum using a cubic kernel. We also estimated the maximum frequency during the $\beta$-bursts. Because the peak frequency of beta oscillation bursts was found to be between 15-20$Hz$, a narrower band ($15 - 20 Hz$) was used to define a $\beta$ oscillation burst for an accurate description of the threshold.

### Estimation of excitation-inhibition balance

The E-I balance a GPe neuron was calculated as the ratio of effective excitatory input it received from the STN neurons ($J_{\text{EI-eff}}$) and effective inhibitory input it received from other

GPe neurons ($J_{\text{II-eff}}$). The effective synaptic weights $J_{\text{EI-eff}}$, $J_{\text{II-eff}}$ were estimated as:

$$J_{\text{EI-eff}} = R_{stn} \times J_{stn-gpe} \times \epsilon_{stn-gpe} \times N_{stn} \times \tau_{exc}$$

where $R_{stn}$ is the average firing rate of the STN neurons, $J_{stn-gpe}$ is the synaptic strength of STN→GPe connection, $\epsilon_{stn-gpe}$ is the probability connection from STN to GPe, $N_{stn}$ is the number of STN neurons and $\tau_{exc}$ is the time constant of the excitatory synapses (Table 1). Similarly, the $J_{\text{II-eff}}$ was estimated as:

$$J_{\text{II-eff}} = R_{gpe} \times J_{gpe-gpe} \times \epsilon_{gpe-gpe} \times N_{gpe} \times \tau_{inh}$$

where $R_{gpe}$ is the average firing rate of the GPe neurons, $J_{gpe-gpe}$ is the synaptic strength of GPE→GPe connection, $\epsilon_{gpe-gpe}$ is the probability connection from GPe to GPe, $N_{gpe}$ is the number of GPe neurons and $\tau_{inh}$ is the time constant of the inhibitory synapse (Table 1).

### Simulation and data analysis tools

The dynamics of STN-GPe network was simulated using NEST (version 2.12.0) [53] with a simulation resolution of 0.1$ms$. The SSB neuron model was added to NEST and the code as well as instructions on recompilation are described in https://github.com/jyotikab/stn_gpe_ssbn. Spiking activity of the network was analyzed using custom code written using SciPy and NumPy libraries. Visualizations were done using Matplotlib [61].

## Results

Beta band (15-30 Hz) oscillations are a characteristic feature of the neuronal activity in PD patients. Animal models have shown that the emergence of $\beta$ band oscillations is also accompanied by a change in the firing rate and spike bursting in both STN and GPe neurons. Here we investigate the effect of firing rate changes and spike bursting in STN and GPe neurons on the power and duration of $\beta$ band oscillations. To this end, we have studied the dynamics of the STN-GPe networks by systematically and independently varying the input firing rate and spike bursting of STN and GPe neurons.

### STN firing rate determines the strength of $\beta$ band oscillations

First, we studied the effect of STN and GPe firing rates on the emergence of oscillations. To this end, we systematically varied the rate of external input to STN and GPe neurons to obtain different firing rates in these neurons and measured the spectral entropy of the population activity to characterize the oscillations (Fig 2). As expected the GPe firing rates monotonically increased as we increased excitatory input to the STN (Fig 2A). However, GPe firing rate varied in a non-monotonic fashion as we increased excitatory input to the GPe neurons (Fig 2A), because of the recurrent inhibition within the GPe. By contrast, STN firing rates monotonically increased as we increased the excitatory input to STN and monotonically decreased as we increased excitatory input to GPe (Fig 2B).

Irrespective of the differences in their mean firing rate, both STN and GPe showed the same oscillation dynamics. More specifically, an increase in the excitatory input to STN or decrease in the excitation to GPe led to the emergence of $\beta$ band oscillations in the STN-GPe network (Fig 2C and 2D—lighter color represents an oscillatory regime). This is consistent with previous studies which showed that increase in excitatory inputs to STN and inhibition to GPe from upstream brain areas (e.g striatum) are sufficient to trigger oscillations in the subthalamo-pallidal circuitry [29, 35].

To characterize the robustness of these results we simulated 10000 networks for which each parameter (connection probability, synaptic strength and delays) were drawn from a Gaussian distribution (see Methods). We estimated the state space of the network activity (characterized by spectral entropy) as a function of external input to STN and GPe. Only the networks which had linearly separable oscillatory and non-oscillatory regions (akin to the Fig 2C and 2D) were selected. The distribution of these selected model parameters closely matched with the parameter sampling distribution (see S2 Fig) however, there were some notable exceptions. For instance, the distributions of the $\epsilon_{stn \rightarrow gpe}$ and $J_{stn \rightarrow gpe}$ were skewed towards lower values for the selected models than the sampled distributions (see S2 Fig). The median, 25% and 75% quartiles of the distribution are displayed in parenthesis (Table 2) beside the original values used in [35]. Overall this robustness analysis suggests that our results are robust for the parameters distributions as shown in the S2 Fig. Fig 3 is reproduced for an exemplary parameter combination (S3 Fig).

These results (Fig 2A–2D) also revealed how the $\beta$ band oscillations depend on the firing rate of the STN and GPe neurons as opposed to change in monotonically increasing input drives. To better visualize this relationship we rendered spectral entropy of the network activity as a function of STN and GPe firing rates (Fig 2E). We found that GPe firing rates are not predictive of the oscillations in the network. For instance, even if GPe firing rate is kept constant, an increase in firing rate of STN neurons was sufficient to induce oscillations. Similarly, a decrease in STN activity reduced oscillations provided GPe firing rates did not vary. On the other hand, when STN firing rate was low (below 5 Hz), any change in the GPe firing rate was not able to induce oscillations. This can also be observed in a scatter plot of spectral entropy against the STN and GPe firing rates (S4 Fig).

We also analyzed the spectrograms of the network activity in three exemplary activity regimes: oscillatory, non-oscillatory and transition regimes (marked as 1, 3 and 2 respectively in Fig 3A). These spectrograms are shown in S5 Fig. The non-oscillatory network (3) showed no oscillations (S5 Fig -top) whereas the oscillatory network (1) showed persistent oscillations (S5 Fig -bottom). The network operating in the transition regime (2) however, showed a propensity towards $\beta$-oscillation bursts even though the oscillations were weak (S5 Fig-middle).

Experimental data [12, 19] as well as previous computational models [29, 31] have suggested that emergence of $\beta$ band oscillations is accompanied by a decrease in the firing rate of GPe neurons and an increase in the firing rate of STN neurons. Our results suggest that only the STN firing rates are positively correlated with the power of $\beta$ band oscillations. Based on these observations we argue that a decrease in GPe activity as observed experimentally may be necessary but not sufficient condition to induce Parkinsonism. That is, reduction in the firing rate of GPe neurons or lesions of GPe are not sufficient to induce beta band oscillations. This suggestion is consistent with the experimental findings that GPe lesions in non-MPTP monkeys do not induce any discernible motor signs of PD [17]. The STN firing rates being predictive of oscillations is also suggested by observations in MPTP treated monkeys [8], where they show that treating STN with muscimol (blocking STN shows decrease in STN firing rates) and intrapallidal blocking of glutamergic receptors (decreased GPe firing rates) suppressed oscillations whereas intrapallidal blocking of GABAergic receptors (increased GPe firing rates) had no effect on the oscillations (measured as power in the beta band).

## State dependent effect of spike bursting neurons on $\beta$ band oscillations

**Effect of spike bursting in GPe neurons on $\beta$ band oscillations.**   Besides changes in average firing rate, dopamine depleted animals also show an increase in spike bursting, in both STN and GPe [8, 49]. Thus far it is not clear whether and how spike bursts affect the $\beta$ band

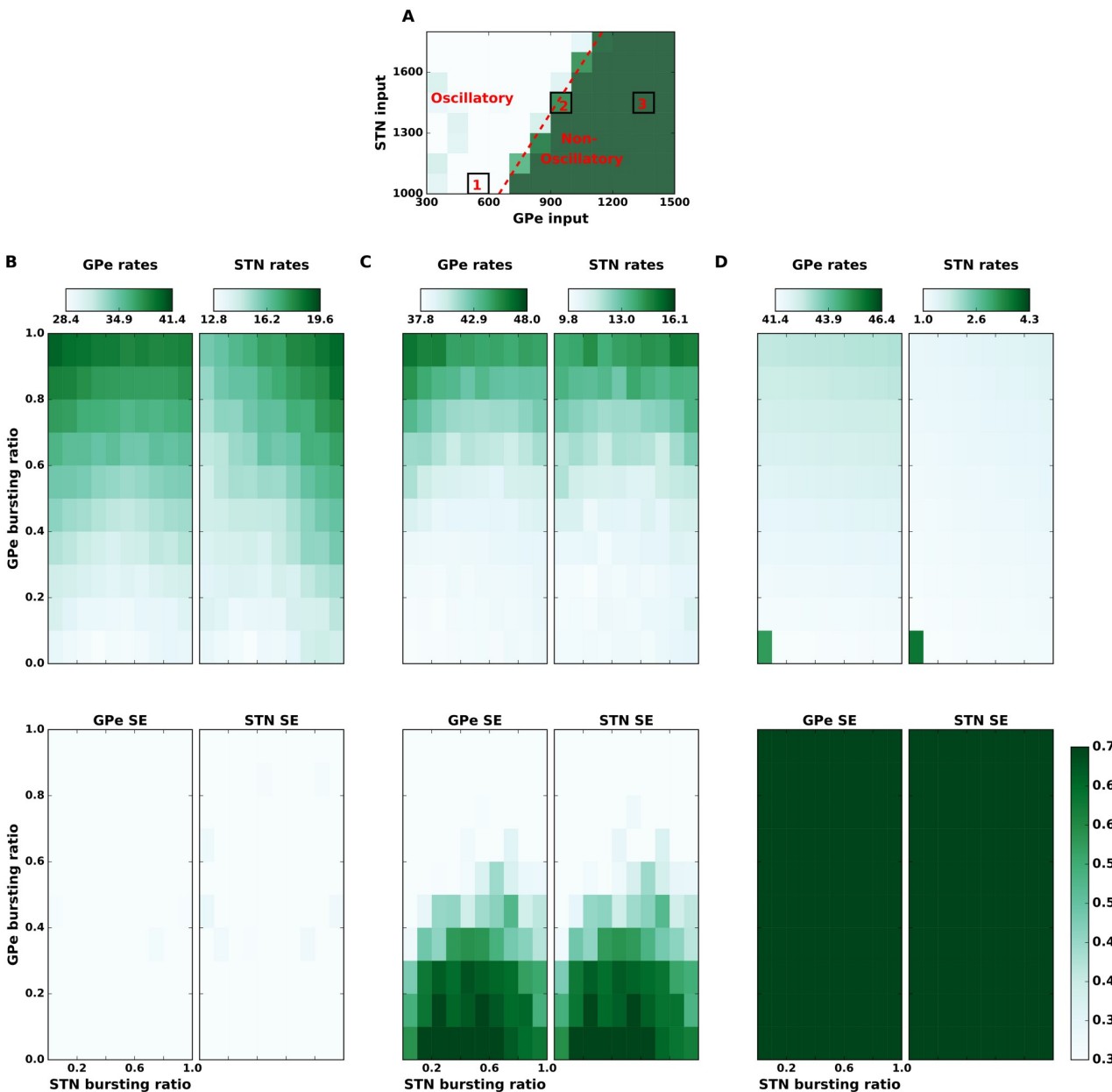

**Fig 3. State dependent effect of spike bursting on the strength of β band oscillations. (A)** Spectral entropy as a function of input to the STN and GPe neurons. This panel is same as the Fig 2C with three regimes of network activity marked as, **1**: oscillatory, **2**: transition regime, **3**: non-oscillatory regime. **(B): Top** GPe (left) and STN (right) firing rates as a function of the fraction of bursting neurons in the STN (x-axis) and GPe (y-axis), in the oscillatory regime **1**. **(B): Bottom** GPe (left) and STN (right) spectral entropy as a function of the fraction of bursting neurons in the STN (x-axis) and GPe (y-axis), in the oscillatory regime (i.e. state **1 in the panel A**). Spike bursting has no effect on the network activity dynamics in this regime. **(C)** Same as in the panel **(B)** but when the network was operating in the transition regime (**marked as** state **2 in the panel A**). In this regime, spike bursting affects the network activity state: increase in the fraction of bursting neurons in GPe induces oscillations whereas an optimal fraction of bursting neurons in STN can quench oscillations. **(D)** Same as in the panel **(B)** but when the network was operating in a non-oscillatory regime (marked as **3** in panel **A**). Addition of BS neurons did not affect a strong non-oscillatory regime.

oscillations. In both reduced or biophysical neuron models introduction of spike bursting necessarily affects the total spike rate of the neuron. As we have shown in the previous section firing rate itself has an effect on the oscillations. That is, such neuron models cannot be used to isolate the contribution of spike bursting on oscillations. Therefore, we used the SSBN model

which allows us to introduce spike bursting in a neuron without affecting its average firing rate [52]. Using this model we systematically altered the fraction of bursting neurons in the STN ($FB_{STN}$) and GPe ($FB_{GPe}$). Previously, in a model of neocortical networks we showed that the effect of spike bursting depends on the network activity states [52]. Therefore, we studied the effect of spike bursting on three exemplary network regimes (1) a strong oscillatory regime, (2) at the border between oscillatory and non-oscillatory regimes (transition regime) and (3) a non-oscillatory regime (marked as 1, 2 and 3 in Fig 3A).

We found that when network was in a strong oscillatory regime (1), an increase in the fraction of bursting neurons in GPe ($FB_{GPe}$) while altered the average firing rates (Fig 3B—upper panel), it had no qualitative effect on the population oscillations (Fig 3B—lower panel). Similarly, when the network was in a non-oscillatory regime (network activity regime 3), $FB_{GPe}$ had no effect on the spike rates and spectrum of the population activity (Fig 3D). That is, in strong oscillatory and completely non-oscillatory states, spike bursting has no consequence for the population activity dynamics.

However, when the network was in the transition regime (network activity regime 2), increase in $FB_{GPe}$ increased oscillations (Fig 3C—lower panel). This activity regime was characterized by weak oscillations when all neurons were non-bursty (S5 Fig-middle panel), but an introduction of spike bursting in $\geq 20\%$ GPe neurons was sufficient to induce oscillations in the STN-GPe network (Fig 3C—lower panel). In this network state, an increase in the number of bursting neurons also increased the average population firing rate (Fig 3C—upper panel) in both STN and GPe. Clearly, this increase in firing rates is a network phenomenon induced by spike bursting and not because of a change in the input excitation (as was shown in Fig 2) or change in the excitability of individual neurons. Finally, an increase in $FB_{GPe}$ increased the network oscillations irrespective of the fraction of bursting neurons in the STN (Fig 3C—lower panel).

In order to ensure that this effect was not dependent on the choice of within burst inter-spike-interval ($B_{isi}$ = 5$ms$), we also measured the effect of spike bursts by changing $B_{isi}$ to 3 ms or 7 ms (S6 and S7 Figs). Qualitatively the effect of spike bursts was not dependent on the $B_{isi}$ however, for smaller values of $B_{isi}$ (3 ms), the region of non-oscillatory regime was reduced. This can also be seen in the corresponding figure showing spectral entropy (S6C Fig—lower panel). By contrast, for higher value of $B_{isi}$ (7 ms), the region of oscillatory regime was reduced (S7B Fig—lower panel).

**Effect of spike bursting in STN neurons on $\beta$ band oscillations.** In contrast to the bursting in GPe neurons, the effect of spike bursting in STN neurons was not only dependent on the network state but also on the fraction of spike bursting neurons in the GPe. Similar to the effect of spike bursting in GPe neurons, in strong oscillatory and non-oscillatory states a change in the fraction of bursting neurons in the STN population had no effect on the network activity state (Fig 3B and 3D, S8 and S10 Figs).

However, in the transition regime (network activity regime 2) spike bursts in the STN affect the oscillations in a non-monotonic fashion. As shown above in this regime an increase in fraction of bursty neurons in GPe pushes the network state towards oscillations. We found that in this regime, the impact of STN spike bursting on oscillations depended on $FB_{GPe}$. For small $FB_{GPe}$, the network remained in a non-oscillatory state and a change in $FB_{STN}$ had no effect on the spectrum of network activity. Similarly, for high $FB_{GPe}$, the network remained in an oscillatory state and a change in $FB_{STN}$ had no effect on the spectrum population activity.

At a moderate fraction of spike bursting neurons in GPe ($0.2 < FB_{GPe} < 0.6$), when the network showed weak oscillations, a small increase in the $FB_{STN}$ reduced oscillations ($FB_{STN} < 0.6$—Fig 3C; S9 Fig) but large values of $FB_{STN}(\geq 0.6)$ enhanced oscillations (Fig 3C). That is, there is a range of parameters for which oscillations enhanced by $FB_{GPe}$ can be quenched by

increasing $FB_{STN}$. As $FB_{GPe}$ increased, more $FB_{STN}$ was required to quench the oscillations and as our results show, beyond a certain point increasing $FB_{STN}$ also leads to persistent oscillations. That is, spike bursting in the STN can suppress or enhance oscillations depending on the fraction of bursting neurons in the GPe.

The non-monotonic effect of STN spike bursting on STN-GPe network oscillation can be better observed in the spectrogram of the population activity of the network (Fig 4). As a fraction of GPe neurons ($FB_{GPe}$ = 40% in this case) were changed to elicit spike bursts (at 1500 ms) $\beta$ band oscillations emerged (Fig 4). These oscillations were quenched when STN neurons also started to spike in bursts from time 3500 ms. When $\approx$ 50% of STN neurons were bursty, the oscillations were almost completely quenched. Any further increase in $FB_{STN}$, however, led to re-emergence of oscillations, albeit at lower frequencies ($\approx$ 15Hz).

Why does $FB_{STN}$ has a non-monotonic effect on the STN-GPe oscillations? The spectrograms of the network activity (Fig 4) revealed that spike bursting in GPe and STN induces oscillations at slightly different frequencies. When $FB_{GPe}$ = 40% and $FB_{STN}$ = 0, the network oscillates at $\approx$20 Hz (1st panel of Fig 4). By contrast, when $FB_{GPe}$ = 40% and $FB_{STN}$ = 100%, the network oscillates $\approx$ 15 Hz (last panel of Fig 4). We hypothesized that the interference of these two oscillations may underlie the non-monotonic effect of spike bursting in STN on $\beta$ band oscillations. For small values of $FB_{STN}$, the two oscillations interfere and generate network activity resembling 'beats', which are reflected as short bursts of $\beta$ band oscillations. These $\beta$ bursts can be observed in the single trial spectrograms—https://osf.io/quycb/—Figures/Spectrograms_single_trials_for_Fig4. It was these short $\beta$ oscillations bursts that resulted in a decrease in overall power in the beta-band (and higher spectral entropy). However, for higher $FB_{STN}$, slower frequency oscillations (generated by STN spike bursting) become strong enough to overcome the GPe spike bursting induced oscillations. To verify our hypothesis we imposed a lower frequency (15 Hz) oscillation on a fraction of STN neurons instead of making them bursty. As we increased the fraction of neurons that oscillated at 15 Hz we observed non-monotonic change in the network oscillation power (S11 Fig). These results are qualitatively similar to those observed when we varied the fraction of bursting neurons in the STN (Fig 4), and provide support to our hypothesis.

Our results show that when the network is operating in the transition regime (network activity regime 2), change in the fraction of spike bursting neurons can control the emergence of $\beta$ oscillations. It is interesting to note that in this regime, the firing rate of STN and GPe neurons falls within the range recorded experimentally (that is, 37–48 spks/s for GPe, 9–16 spks/s for STN) for healthy conditions. This also suggests that in healthy states, GPe-STN network may be operating in the regime at the border of oscillatory and non-oscillatory state. In this regime, spike bursting may provide an additional mechanism to generate short lived $\beta$ band oscillations ($\beta$-oscillation bursts) as has been observed in healthy rats [45], that is, an increase in spike bursting in the GPe can induce oscillations, which can be quenched provided STN neurons also elicit spikes in bursts.

## Comparison of firing rate changes due to spike bursting and input drive

When neurons do not spike in bursts, only STN firing rate is predictive of $\beta$ band oscillations (see Fig 2E). While in our neuron model (SSBN) spike burst and neuron average firing rates can be independently varied, spike bursts may change the network activity state and thereby affect the firing rates of STN and GPe neurons. However, with this neuron model, we can isolate the changes in STN and GPe firing rates purely due to the effect of spike bursting on the network activity (once the input drives are fixed). Therefore, we estimated how the firing rates of STN and GPe neurons affect the $\beta$ band oscillations when neurons are allowed to spike in

**Mean spectrogram over 5 trials**

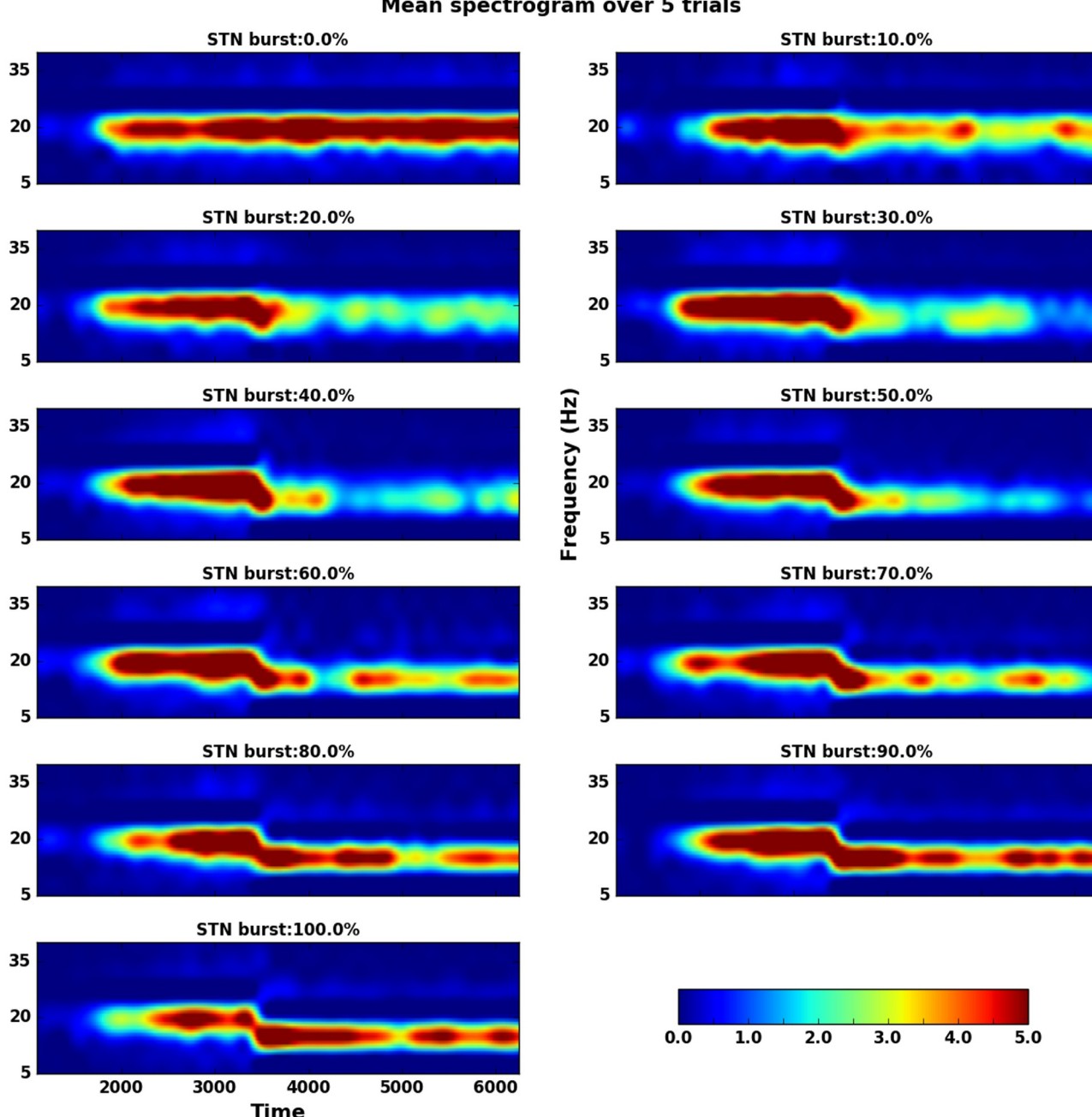

**Fig 4. Non-monotonic effect of STN spike bursting on network oscillations when the network operates in the transition regime.** Here the fraction of bursting neurons in the GPe was fixed to 40% of GPe neurons and the fraction of bursting neurons in the STN ($FB_{STN}$) was increased systematically (as marked on different subplots). 40% of GPe neurons were made to elicit spike bursts from time point 1500 ms. This resulted in emergence of oscillations. A fraction of STN neurons ($FB_{STN}$ marked on each subplot) were made to burst, starting at time 3500ms. For small to moderate $FB_{STN}$, oscillations disappeared. But when $FB_{STN}$ was larger oscillations reappeared albeit at a lower frequency. The spectrograms shown here were averaged over 5 trials of the network with different random seeds.

bursts.To this end, we fixed the input rates such that the network operated in one of the six representative activity states (marked as 1, 1′, 2, 2′, 3, 3′ in Fig 5) and systematically varied the fraction of bursting neurons in the STN and GPe. At each operating point, spike bursting resulted in a change in the average firing rate of the neurons because spike bursting perturbed

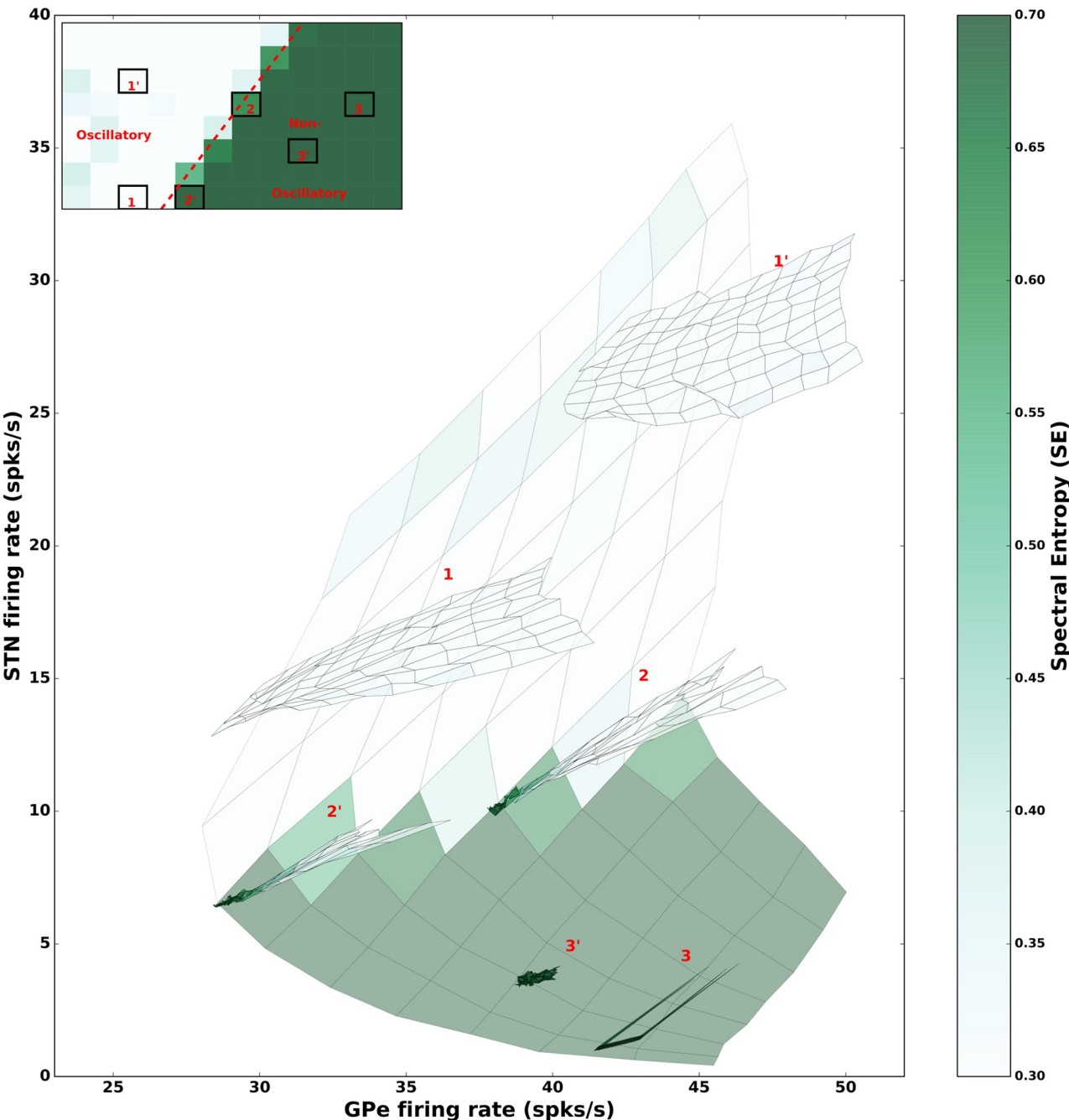

**Fig 5. Comparison of firing rate changes induced by the input drive and spike bursting.** The pale background (same as in Fig 2E) illustrates the effect of GPe and STN firing rates on oscillations. This is used to compare the effects of firing rates and spike bursting. The inset shows the 6 network states chosen for the comparison: two oscillatory (1 and 1′), two border (2 and 2′) and two non-oscillatory (3 and 3′). In each of the chosen states, we varied the fraction of bursting neurons in both STN and GPe populations from 0 to 100%. For each combination of the fraction of spike bursting neurons we estimated the firing rate of STN and GPe neurons and their corresponding spectral entropy. Then firing rates and spectral entropy are plotted to create the six manifolds. The size of manifolds is much smaller than the background indicating that the changes in firing rates induced solely by spike bursting is rather small.

the network operating point (Fig 5). We found that in non-oscillatory states (3, 3′) spike bursting had a very little effect on the average firing rate of the neurons and on the network activity state (Fig 5 green region). By contrast, in oscillatory states (1, 1′) spike bursting resulted in relatively large change in the firing rates (Fig 5 white region). In both regimes, the change in firing rates due to spike bursting were not sufficient to change the network state qualitatively. In the transition regime (e.g. network states 2, 2′) spike bursts led to higher firing rate in STN and GPe neurons. Interestingly, in the transition regime, with spike bursting, an increase in the firing rate of both GPe and STN neurons lead to increase in oscillations. (Fig 5 border between white and green regions).

Lastly, this analysis is consistent with the previous observation that only in the transition regime, does spike bursting qualitatively change the network state. The location of the network in the STN-GPe rate state space is determined by the external input drives and only in the transition regime (2, 2′), spike bursting changes the network state from non-oscillatory to oscillatory with an increase in STN and GPe firing rates.

## Control of the amplitude and duration of $\beta$ band oscillation bursts by spike bursting

Next, we explored how the proportion of GPe and STN spike bursting neurons affects the amplitude and duration of $\beta$ oscillation bursts. In particular we were interested in identifying the fraction of spike bursting neurons needed to obtain $\beta$ oscillation bursts similar to those recorded in the BG during healthy conditions. The length of a $\beta$ oscillation burst was defined as the duration that the beta band amplitude envelope remained above the threshold (Fig 6A). The threshold (Fig 6A and 6B) was defined as the averaged maximum (over 5 trials) of the $\beta$ band amplitude estimated for an ensemble of Poisson type spike trains with the same firing rate as that of our network activity. Note that these $\beta$ oscillation bursts were calculated on the amplitude envelopes of the individual trials and not on the average amplitude envelope corresponding to the spectrogram shown in Fig 4. The spectrograms of the individual trials can be found on the OSF project—https://osf.io/quycb/—Figures/Spectrograms_single_trials_for_Fig4.

An increase in the fraction of spike bursting neuron in the GPe increased the average $\beta$ oscillation burst length. However, an increase in the STN spike bursting ratio had a non-monotonic effect on the beta oscillation burst length as we expected given the effect of $FB_{STN}$ on spectral entropy (Fig 3C). This also suggests a positive correlation between the $\beta$ oscillation burst length/duration and measured power in the $\beta$ band in monkeys [42] and PD patients [46]. However note that in [46], the $\beta$ oscillation burst rate and amplitude are better correlated with $\beta$ band power than the $\beta$ oscillation burst length. The $\beta$ oscillation burst amplitude, however, increased with an increase in fraction of bursting neuron in both GPe and STN (Fig 6D).

To compare the model output with the experimental data for rodents we measured three features of the network activity for all combinations of $FB_{STN}$ and $FB_{GPe}$: average $\beta$-oscillation burst length, average $\beta$-oscillation burst peak frequency, and correlation between $\beta$-oscillation burst length and amplitude. The average $\beta$-oscillation burst length measured in healthy mice is $\approx 0.2$ s [62]. The $\beta$-oscillation burst duration and $\beta$-oscillation burst amplitude in humans [20] as well as non-human primates [63] is positively correlated that is, stronger oscillatory bursts also last longer. Recent data also suggests a positive correlation between beta amplitude and duration in 6-OHDA lesioned rats, however it is stronger in GPe as compared to STN [64]. The presence of such a relationship in healthy rats is not explored and therefore remains an assumption of our model.

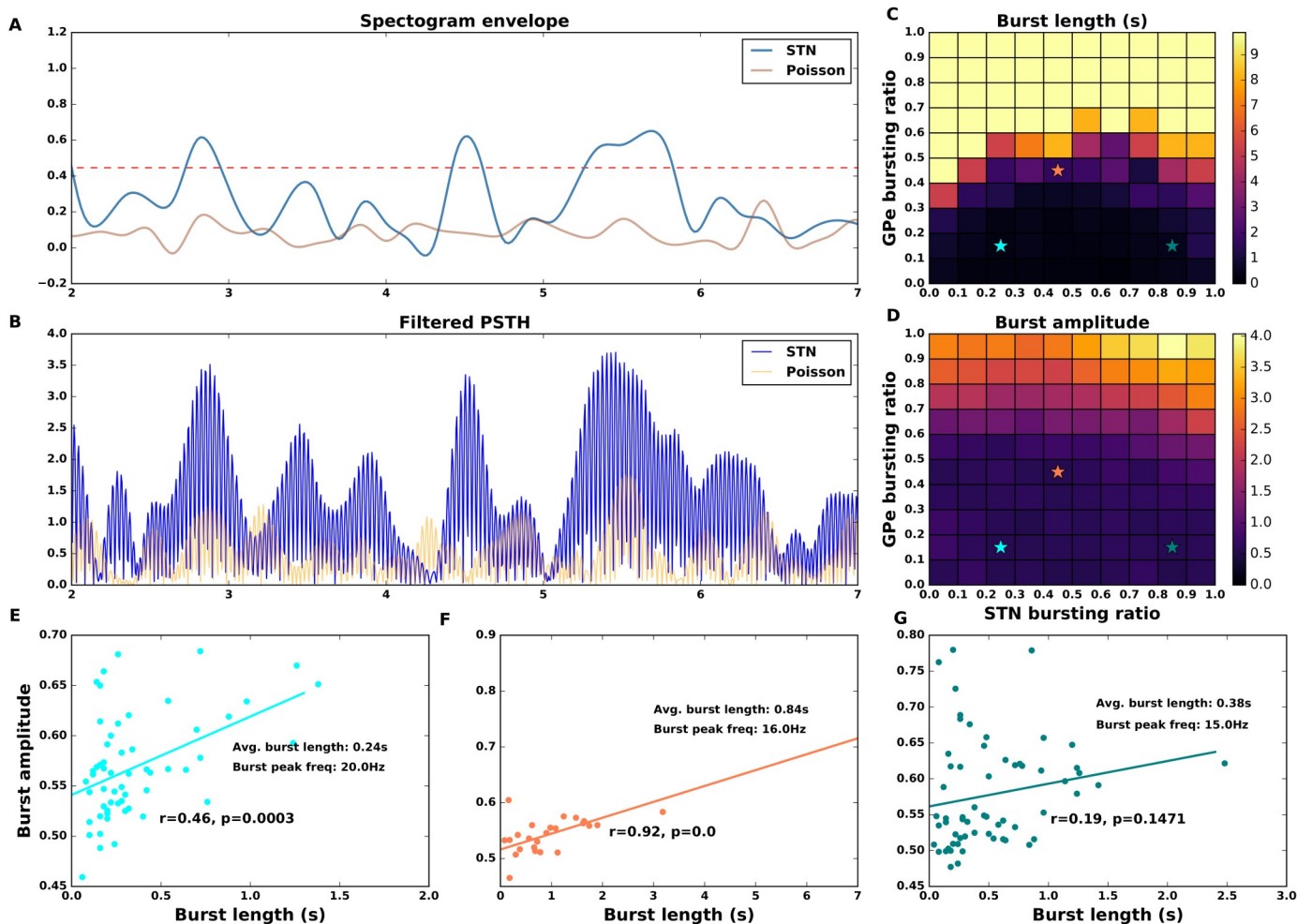

**Fig 6. Effect of spike bursting on beta-band oscillation bursts. (A)** An example of the amplitude envelope of the beta band (15-20 Hz) oscillations (blue trace). Beta oscillation burst threshold (red dashed line) was determined by averaging the maximum of beta band amplitude envelop for a Poisson process (orange trace) with the same firing rate as the neuron in the STN-GPe network. The averaging was done over Poissonian firing rates corresponding to all GPe and STN spike bursting ratios and 5 trials per STN-GPe bursty ratio combination. **(B)** Low pass filtered (15-20 Hz band) trace of population firing rate in the STN population in the beta band (15-20Hz). The orange trace shows the population firing rate of the Poisson process with same average firing rate as the STN activity. **(C)** Beta oscillation burst length as a function of the fraction of spike bursting neurons in the GPe and the STN. **(D)** Beta oscillation burst amplitude as a function of the fraction of spike bursting neurons in the GPe and the STN. **(E,F,G)** Correlation between $\beta$ oscillation burst length and amplitude for three different combinations of $FB_{STN}$ and $FB_{GPe}$(marked with cyan, orange and green colors in the pane **C**. Cyan marker shows beta oscillation burst length and amplitude for 10% of spike bursting neurons in GPe and 20% in STN—this combination of spike bursting neurons gives an average oscillation burst length of 0.24 s which is comparable to experimentally measured values. In panels **E-F** the p-values are listed to 4 places after decimal point.

According to these measures, the regime with a small fraction of bursty neurons in GPe (e.g. 10%) and STN (e.g. 20%) (Fig 6C and 6D—cyan marker) resembled most closely with the experimentally measured values of all the aforementioned features. In this regime, the oscillation burst peak frequency was $\approx$ 20Hz. Moreover, oscillation burst amplitude and oscillation burst length (mean value:$\approx$ 0.24 s) were positively correlated ($r_{bl,ba}$ = 0.46, $p \leq 0.0002$) (Fig 6E).

For a higher fraction of spike bursting neurons in GPe (40%) and STN (40%—Fig 6F), the average $\beta$-oscillation burst lengths increases to $\approx$ 0.8s, the intraburst frequency decreases to $\approx$ 16Hz and the positive correlation between $\beta$-oscillation burst amplitude and $\beta$-oscillation burst length is high and significant ($r_{bl,ba}$ = 0.92, $p < 0.0001$). In a regime with a lower fraction

of spike bursty neurons for GPe (10%) and a higher fraction of spike bursting neurons in STN (80%), the positive correlation between $\beta$-oscillation burst length and $\beta$-oscillation burst amplitude was not significant, however the $\beta$-oscillation burst length is slightly higher ($\approx 0.4$ s) and $\beta$-oscillation burst peak frequency is slower ($\approx 15$Hz).

The $\beta$-bursts are a population phenomenon in our model. To test whether $\beta$-bursts are also observed in individual neurons we estimated the spectra of individual neuron firing rates (S12 Fig). As expected, the $\beta$ oscillation bursts were more prevalent in neurons with higher firing rates. Moreover, it was not necessary for a neuron to spike in all $\beta$-oscillation bursts, which indicates that the these oscillatory bursts are a network effect. Furthermore, the oscillation frequency for the single neuron was more variable than the population frequency at $\beta$ (S12 Fig -GPe bursty #3, STN bursty #3).

Based on these results, we predict that short lived $\beta$-burst in healthy mice are generated when $\approx 10\%$ of GPe neurons and $\approx 20\%$ of STN neurons elicit spike bursts.

It is unclear how these neurons are entrained to produce spike bursts. The spike bursts in the GPe neurons could be caused by spike bursts in striatal neurons [64, 65] in rats. In 6-OHDA lesioned rats, with an increase in number of spikes/burst in striatal neurons, GPe neurons show an increased burst index [65]. Striatal inactivation (by muscimol) significantly decreased such spike bursts in the GPe [65]. This could be a result of both alleviating changes in the operating point of STN-GPe network as result of striatal inactivation (increased GPe firing rates [65]) or/and the lack of entrainment of GPe neurons into bursts. The GPe bursting could in turn entrain STN neurons to burst with large bouts of synchronized inhibition, which has been also suggested by [64] because the phase of prototypical GPe neurons leads that of STN neurons by a cycle. On the other hand, STN and GPe could also be entrained simultaneously by cortical $\beta$-bursts directly or via striatum respectively [64]. Besides, these network interactions, spike bursts could be caused by changes in the neuron properties due to lack of dopamine or appropriate inputs. This is consistent with increased spike bursting in rat STN slices with an increase in hyperpolarization of the neuron's membrane potential [66].

## Dependence of the network states on the excitation and inhibition balance

Finally, to better understand the impact of firing rate changes and spike bursting neurons on the $\beta$ band oscillations we analyzed the balance of effective excitation and inhibition (E-I balance) in the network for different input firing rates and fractions of spike bursting neurons. E-I balance is the primary determinant of oscillations in spiking neuronal networks [40]. To get an estimate of the E-I balance for a GPe neuron we measured effective excitation it received from a STN neuron ($J_{\text{EI-eff}}$) and effective inhibition it received from other GPe neurons ($J_{\text{II-eff}}$). We estimated the effective excitation and inhibition for all combinations of external input as shown in Fig 2 (See Methods).

Consistent with the previous theoretical work on neuronal network dynamics we found that the non-oscillatory states emerged when effective inhibition received by a GPe neuron was much higher than the effective excitatory inputs, whereas oscillatory states appeared when the effective excitation from STN to a GPe neuron increased (Fig 7).

Next we mapped the effect of GPe and STN spike bursting on the E-I balance in the three exemplary network states (1: oscillatory, 2: border of oscillatory and non-oscillatory, 3: non-oscillatory). As expected we found that in the oscillatory state 1, increase in GPe spike bursting increased the effective inhibition and excitation whereas STN spike bursting has a non-monotonic effect (Fig 7—state marked as 1). However, in this state bursting in either population was not strong enough to change the E-I balance in order to introduce a qualitative change in the network state. Similarly for the non-oscillatory state 3, a change in the fraction

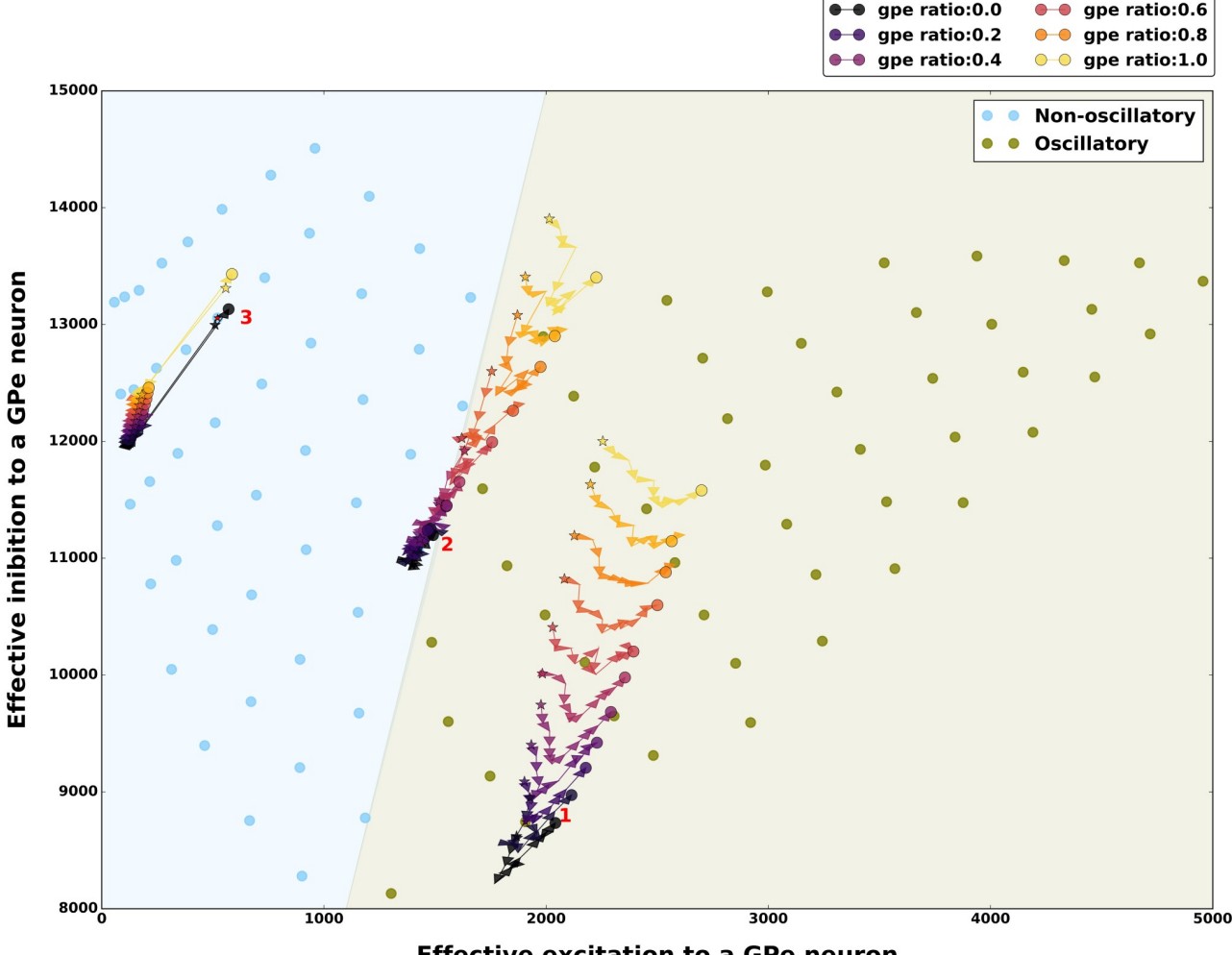

**Fig 7. Effect of spike bursting on the excitation-inhibition balance in different network activity regimes.** E-I balance was characterized by estimating the total effective excitation and inhibition received by a GPe neuron (see Methods). E-I balance for oscillatory and non-oscillatory network states for 100% non-bursting neurons. Each filled circle shows E-I balance for different external inputs to STN and GPe neurons shown in Figs 2 and 3. The effect of spike bursting on E-I balance is shown for the three exemplary network activity regimes: 1-Oscillatory regime, 2-Transition regime, 3-Non-oscillatory regime (see Fig 3 for details). Different colored stars and filled circles show how the E-I balance varied as function of change in the fraction of spike bursting neurons in the GPe (warmer colors indicate higher % of spike bursting neurons). The trajectory from the star (STN spike bursting ratio = 0%) to the filled circle shows change in the E-I balance as the fraction of spike bursting in STN is varied from 0% to 100%. In all the states spike bursting tends to make the network activity more oscillatory, however, the amount by which spike bursting is able to push the network towards oscillatory regime depends on the network activity regime itself.

of spike bursting neuron in the GPe and STN spike bursting was not sufficient to introduce any qualitative change in the state of the network (Fig 7). When the network was in the regime 2, even though increase in fraction of bursting neuron in the GPe introduced a small change in the effective E-I balance, it was sufficient to move the network activity into the oscillatory regime from non-oscillatory regime. Increased in the fraction of bursting neuron in the STN showed a non-monotonic effect on the E-I balance and while a moderate amount of $FB_{STN}$ pushed the network towards the non-oscillatory regime, which was not the case for a higher $FB_{STN}$ (Fig 7).

## Discussion

PD is characterized by change in both firing rate and firing patterns of GPe and STN as shown in animals models [8, 49–51]. In this study, we focused on uncoupling the roles of STN and GPe population firing rate and firing patterns (spike bursting) in determining the presence of oscillations. Our results show that an increase in the firing rate of STN neurons is the primary determinant of oscillations in the STN-GPe network, however the effect of changes in GPe firing rates is contingent on the firing rate of STN neurons. Similarly, the effect of increase in spike bursting in STN and GPe neurons is contingent on the dynamical state of the network.

### Effect of firing rate changes on $\beta$ band oscillations

In our model network, an increase in the firing rate of STN neurons was sufficient to drive the network into an oscillatory state, irrespective of the firing rate of the GPe neurons. By contrast, a decrease in the firing rate of GPe neurons was able to generate $\beta$ band oscillations only when STN neuron firing rate also increased (Fig 2E, S4 Fig).

A change in GPe and STN firing rates also alter the effective excitation-inhibition of the network (Figs 7 and 8). The non-oscillatory network states were observed in the inhibition dominant regime (when effective inhibition to a GPe neuron was higher than effective excitation). An increase in effective excitation altered the regime to oscillatory. This result may explain the experimental observation that the therapeutic effect of DBS in human and non-human primates is accompanied by a corresponding decrease in STN firing rates [67, 68] and an associated corresponding increase in GPe/GPi firing rates [67–69] and thereby tipping the network balance towards the inhibition dominant.

Our results also show that if the firing rate of STN neurons remains fixed, changes in the firing rate of GPe neurons are not sufficient to influence the oscillations. Indeed, it can be argued that because STN and GPe are recurrently connected, their firing rates cannot independently change. However, these results imply that the $\beta$ band oscillations are more sensitive to changes in STN firing rates than to GPe firing rates. This is consistent with the observations that STN inactivation with muscimol (decreased STN firing rates) suppressed oscillations [8] in monkeys. By contrast, intrapallidal blocking of GABAergic receptors (increased GPe firing rates) either had no effect or increased the oscillations [8]. In our model there are two possible mechanisms to induce beta-band oscillations: (a) The *indirect pathway induced* oscillations can be initiated by reducing the firing rate of GPe neurons via transient increase in firing rate of D2-spiny projection neurons. (b) The *hyper-direct pathway induced* oscillations can be initiated by a transient increase in the firing rate of cortical neurons projecting onto the STN neurons. Our results suggest that the *indirect pathway induced* oscillations can be quenched by transiently decreasing the activity of STN neurons but the *hyper-direct pathway induced* oscillations cannot be countered by transiently increase the activity of GPe neurons.

At the behavioral level, the sensitivity of $\beta$ band oscillations to STN firing rates could provide an explanation for the importance of STN in response inhibition in general and, especially when there is an increase in potential responses (high conflict task). Experimental data in humans have shown that the STN firing rates increase in proportion to the degree of conflict in an action-selection task [70]. Interestingly, the increase in STN firing rates during a high conflict task is also accompanied by an increase in $\beta$ band activity [71] and is reminiscent of increase in rat STN activity [72] as well as power of the $\beta$ band oscillations observed in successful STOP trials [73]. Furthermore, the latency [73] as well as amount of modulation [74] in STN $\beta$ band oscillations are correlated with the speed of an action. All these observations suggest there may be a functional rationale to the sensitivity of oscillations to STN firing rates as shown by our results. That is, an increase in STN firing rates could be a mechanism to delay

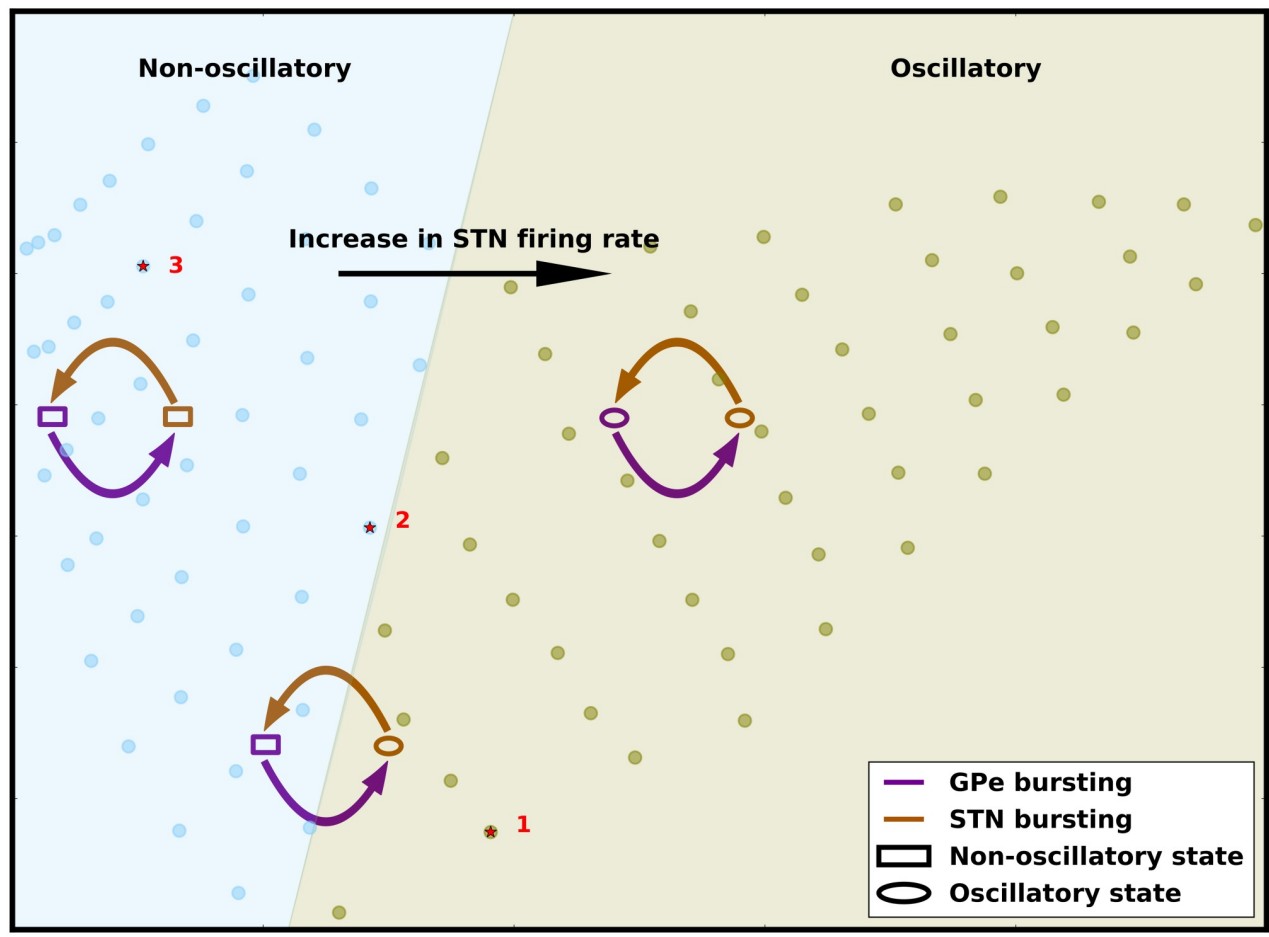

**Fig 8. Summary of the effect of firing rate and spike bursting on network state.** The background image (same as Fig 7) show the oscillatory and non-oscillatory regimes of STN-GPe network as a function of effective excitation and inhibition. The arrows schematically show the change in EI-balances as we increase spike bursting in the STN or GPe. The STN-GPe network oscillations are more sensitive to the STN firing rate. The balance of STN and GPe firing rates determines the global state of network activity. Spike bursting in GPe always increases both effective inhibition and effective excitation. Small increases in spike bursting in STN results in a decrease in both effective excitation and effective inhibition and thereby, reduces oscillations. By contrast, a large increase in the fraction of spike bursting neurons in the STN increases both effective inhibition and effective excitation and thereby, enhances oscillations. However, this effect is smaller and therefore, spike bursting is effective in altering the network oscillations only when the network is operating close to the border of oscillatory and non-oscillatory states.

the decision making ("hold the horses" [75]) by increasing the $\beta$ band activity, which cannot be vetoed by the GPe and thereby plays a vital role at response inhibition [76].

## Effect of changes in spike bursting on beta band oscillations

Our results show that the effect of GPe or STN spike bursting is dependent on the state of the network as defined by the firing rates of the STN and GPe neurons. In a regime with strong oscillations, GPe and STN spike bursting does not qualitatively change the network state and the network remains oscillatory. Similarly, in a non-oscillatory regime, GPe and STN spike bursting has no qualitative effect on the network state. However, in a regime at the border of oscillatory and non-oscillatory, an increase in bursting neurons in the GPe induces oscillations but the effect of increasing STN spike bursting neurons depends on the fraction of GPe spike bursting neurons. In this regime, when spike bursting neurons in the GPe induce oscillations ($0.1 \leq FB_{GPe} \leq 0.4$), a small increase in the fraction of bursting neurons in the STN disrupts the

oscillations. However, a large fraction of spike bursting neurons in the STN re-instate the $\beta$ band oscillation (Fig 4). This non-monotonic effect of spike bursting neurons in the STN is because when neurons spike in bursts both STN and GPe tend to induce oscillations at slightly different frequencies (Fig 4, S11 Fig). The relative power of these oscillations depends on the fraction of spike bursting neurons in the two populations. When $FB_{GPe} = FB_{STN} \approx 0.5$ the magnitude of the two oscillations is comparable and they produce 'beats' resulting in a reduction in the power of $\beta$ band oscillations. However, if $FB_{GPe} \geq 0.5$ or $FB_{STN} \geq 0.5$, the stronger of the two oscillations overcomes the other, resulting in the higher power in the $\beta$ band.

Similar to the rate effect, the effect of spike bursting can also be captured by calculating the balance of effective excitation and inhibition in the network (Figs 7 and 8). GPe bursting increases both the effective excitation and inhibition to a GPe neuron. Therefore, when a network is operating close to the border of oscillatory and non-oscillatory regime, increase in bursting in GPe neurons pushes the network to an oscillatory regime (Fig 8). An increase in spike bursting neurons in the STN, however, has a non-monotonic effect—a small number of bursting neurons counter the effect of GPe bursting by decreasing both effective excitation and inhibition. However the effect of larger number of STN neurons bursting collude with the effect of GPe spike bursting by increasing both effective excitation and inhibition (Fig 8).

During PD, both STN and GPe neuron show an increase in spike bursting activity in monkeys [8, 49] and rats [50, 51]. Based on our results, we propose that increase in STN bursting might play a compensatory role in an attempt to quench the burst induced oscillations as a self-regulating mechanism. However, it has been shown that dopamine depletion itself leads to increased spike bursting in STN slices [77–79].

The effective excitation-inhibition change induced by the striatal/cortical inputs to STN/GPe neurons is much bigger than the change induced by spike bursts. We corroborated this by the observation that the firing rate changes solely due to spike bursting are much smaller than the firing rate changes due to input drives (Fig 5). This is the reason why spike bursts failed to change the network states when it was operating in strongly asynchronous or oscillatory states (Fig 3B and 3D). These modest firing rate changes due to spike bursting, however can change the state of a network operating at the border of the oscillatory and non-oscillatory regime. Thus, based on these results we propose that the change in GPe and STN firing rates determines the underlying network state whereas neuronal spike bursting fine tunes it.

## Tandem of GPe-STN spike bursting generates beta oscillations bursts

In healthy conditions, short epochs of oscillations ($\beta$ bursts) have been observed in rodents [45, 62] and non-human primates [63]. They are also observed in Parkinsonian patients during dopamine ON state [20]. The precise function of $\beta$ bursts in healthy conditions is currently unknown but they tend to occur before movement (e.g after the cue [45]) and disappear when the movement is initiated [80–83]. Beta bursts become longer and stronger during Parkinsonian conditions [20], therefore, they are thought to be correlated with impairment of voluntary movement in PD patients [20, 46, 47]. The average length of the $\beta$ bursts in healthy rodents last for an average of 0.2 sec [62]. In our model, we can generate the oscillatory $\beta$ bursts of average burst length 0.24 s by making 10% of GPe and 20% of STN neurons are of spike bursting type (Fig 6). We propose that an interplay of spike bursts in a STN-GPe network lying on the border of oscillatory and non-oscillatory regime may be the underlying mechanism to generate short bursts of $\beta$ oscillations.

Experimental results in rat brain slices have shown that an increased spike bursting in STN is associated with an increase in hyperpolarization of the neuron's membrane potential [66]. That is, spike bursts in the GPe network (e.g. because of striatal bursts [64]) can induce spike

bursting in the STN neuron by inducing large synchronized inhibition. However, if only less than 50% of the GPe neurons generate spike bursts, an equivalent proportion of neurons bursting in STN will quench the oscillations resulting in a short-lived "$\beta$ burst".

However, in pathological conditions, the network state could be pushed into the oscillatory regime (either due to a change in firing rates or excessive spike bursting) where these oscillations can no longer be quenched. This has been explained in the summary figure (Fig 8).

Our results also suggest that in healthy conditions the network might operate on the boundary of synchronization and asynchronization regime. Operating at the boundary enables the network to make incursions into the oscillatory regime (when GPe neurons elicit spike bursts) and retreat to the asynchronization regime (when STN neurons elicit spike bursts) with a proportion of spike bursting neurons where such self-regulated transitions are possible. However, in pathological conditions, the network very likely shifts deeper into the oscillation regime (due to the change in firing rates or excessive spike bursting), where no amount of STN bursting can push the network back to asynchronized regime. A similar idea was suggested by [84, 85] where they explored the effect of excitatory input drive to GPe ($I_{app}$) and STN-GPe synaptic strength ($g_{syn}$) on $\beta$ band oscillations. They found that a higher drive to GPe ($I_{app}$) and lower STN-GPe synaptic strength ($g_{syn}$) leads to asynchronous activity whereas low input drive to GPe and high STN-GPe synaptic strength leads to a strong oscillatory state. These results are consistent with our observations, that an increased excitatory drive to GPe leads to asynchronous activity and vice versa (Fig 2C and 2D). The regime on the border yields intermittent synchronous states that resembles the experimental data measured from PD patients.

Rubchinsky and colleagues argued that healthy states should also operate on the boundary, as it offers many advantages such as easy creation and dissolution of transient neuronal assemblies [84, 85] as required by functioning of network shown in other parts of basal ganglia (especially striatum, [86–90]). We also propose that the STN-GPe network should operate close to the border between oscillatory and non-oscillatory states because it makes it easy to generate short epochs of $\beta$ band oscillations which are often observed in behaving animals.

## Model limitations

Here we aimed to use a minimal model sufficient to dissociate the effect of firing rates and spike bursts on the dynamics of STN-GPe network. The model was constrained by experimental data on synaptic connectivity and neuronal firing rates in healthy states. However, the model has a number of limitations. For instance, the neuron model that is effective in isolating the effects of firing rates and spike bursts, assumed that in every spike bursts, a fixed number of spikes are elicited. Therefore, our model cannot account for phenomena such as firing rate adaptation within bursts. Furthermore, we have focused on spike burst changes in the STN-GPe network alone. Inputs from other sources such as pallidostriatal [26], thalamocortical or thalamostriatal projections may also influence the $\beta$ oscillation bursts but were not considered in our model.

With regard to the oscillations we specifically focused on the $\beta$ band, however the oscillations in $\beta$-band are known to be closely related to oscillations in other frequency bands, especially $\gamma$-band [26, 59, 91–93]. The analysis should be extended to include other frequency bands and their relation to $\beta$ oscillation bursts. Moreover, we also do not distinguish between high and low frequency $\beta$ bands oscillation. We show that the frequency of oscillations could be determined by the proportion of GPe and STN neurons that are bursty. There is evidence for a drift in oscillation frequency from high to low $\beta$ bands in striatal LFPs during episodes of increased $\beta$ band power triggered by infusion of cholinergic agonist in awake mice [25]. Moreover, it has been suggested that the low $\beta$ band oscillations are anti-kinetic and gets regulated

by dopamine whereas high $\beta$ maybe non-pathological in nature [10] in humans [94], monkeys [95] and rats [59]. Hence the issue of different $\beta$ oscillations bands needs to be investigated in further detail.

In terms of the mechanisms underlying the emergence of $\beta$ band oscillations we have explored only two causes of these oscillations: changes in firing rates and spike bursting, however there may be various other factors that can modulate the $\beta$ band oscillations. Indeed, the BG network has multiple excitatory-inhibitory loops capable of inducing oscillations. We have also assumed that input the STN-GPe neurons is aperiodic, uncorrelated Poisson distributed spike trains. This choice was made to explore the response of the network to firing rate changes in the input drive. However, inputs to STN-GPe are richer in their statistics and dynamics, e.g. bursty, periodic, correlated [24, 30, 35, 64, 96, 97]. Such non-Poissonian inputs might underlie resonance of STN-GPe network at certain frequencies [34]. In future models effect of non-Poissonian inputs should explored in more detail.

As is typical for computational models, necessary parameters are rarely available from a single animal model and single experimental conditions. To counter this limitation, we varied the parameters by 10-20% to ensure the robustness of our results. Even though the model was constrained by data from rodents some of the model results are consistent with experimental observations made in non-human primates and human patients. This similarity underscores the generality of the model and the experimental phenomena (i.e. properties of beta band oscillations).

## Conclusions

Despite the simplicity of our model, our analysis of the STN-GPe network provides new insights about the role of spike rates, spike bursts and varied roles of STN and GPe in shaping of the dynamics of beta band oscillations suggest several means of quenching the pathological oscillations for instance by (1) reducing the firing rate of the STN neurons, (2) reducing the excitability of STN neurons, and (3) by balancing the fraction of bursting and non-bursting neurons in the STN and GPe.

## Supporting information

**S1 Fig. State dependent Stochastic Bursting Neuron (SSBN) model. (A)** Membrane potential and spiking pattern for different number of spikes per burst.**(B)** Input current and output firing rate ($f - I$) curve of the SSBN for different number of spikes per burst.
(TIFF)

**S2 Fig. Robustness analysis.** The areas in gray color of the violin plot shows the distribution that was sampled for robustness analysis. The areas in brown color of the violin plots show the distribution of the parameters that qualitatively reproduce the key results shown in the Fig 3A. See Methods for more detail.
(TIFF)

**S3 Fig. Effect of spike bursting on STN-GPe network oscillations for an example network from robustness analysis. (A)** Spectral entropy as a function of input to the STN and GPe neurons for a different set of model parameters than used in Fig 3. In this panel the location of the red dotted line and the three exemplary activity regimes are marked in the same place as in Fig 3A for the ease of comparison. **(B): Top** Same as Fig 3B-top. GPe (left) and STN (right) firing rates as a function of the fraction of spike bursting neurons in the STN (x-axis) and GPe (y-axis), in the regime 1. **(B): Bottom** Same as Fig 3B-bottom in the main text. However note that, in this network, regime 1 is on the border and hence shows the non-monotonic effect of

STN spike bursting on oscillations as observed in Fig 3C-bottom. **(C)** Same as Fig 3C in the main text. However note that in this network regime 2 is deeper into non-oscillatory regime as compared to Same as Fig 3C. Hence, the effect of spike bursting on oscillations is close to being ineffective. **(D)** Same as in the panel Fig 3D in the main text. These results are qualitatively similar to the ones shown in the Fig 3. Here, however we have used a different set of parameters than the Fig 3 ($J_{gpe-gpe}$ = -0.67, $J_{gpe-stn}$ = -1.0, $J_{stn-gpe}$ = 1.04, $\epsilon_{gpe-gpe}$ = 0.02, $\epsilon_{stn-gpe}$ = 0.02, $\epsilon_{gpe-stn}$ = 0.03, $\tau_{stn-gpe}$ = 5.96$ms$, $\tau_{gpe-gpe}$ = 3.14$ms$, $\tau_{gpe-stn}$ = 5.34$ms$.
(TIFF)

**S4 Fig. Spectral entropy as function of GPe (Top) and STN (Bottom) firing rates. Different colors indicate five different trials with same parameters.** Different dots correspond to network simulations with different parameters. For a wide range of GPe firing rates the network can be in an oscillatory or non-oscillatory states, however, high firing rate in STN is necessary to induce oscillations.
(TIFF)

**S5 Fig. Spectrograms of network activity in three exemplary network activity regimes. top**: Non-oscillatory regime (marked as 3 in Fig 3A). **middle**: Transition regime (marked as 2 in Fig 3A). **bottom**: Oscillatory regime (marked as 1 in Fig 3A in the main text).
(TIFF)

**S6 Fig. Reproduction of results shown in Fig 3 for a smaller intra-burst inter-spike-interval ($B_{isi}$ = 3$ms$).** The positions of the regimes 1, 2 and 3 as well as the dashed line dividing the oscillatory and non-oscillatory regime are kept same as in Fig 3A in the main text. Decreasing the within burst inter-spike-interval resulted in reduction in the area of non-oscillatory regime.
(TIFF)

**S7 Fig. Reproduction of results shown in Fig 3 for a larger intra-burst inter-spike-interval ($B_{isi}$ = 7$ms$).** The positions of the regimes 1, 2 and 3 as well as the dashed line dividing the oscillatory and non-oscillatory regime are kept same as in Fig 3A. Increasing the within burst inter-spike-interval reduced the region of the oscillatory regime.
(TIFF)

**S8 Fig. Effect of spike bursting when the network was operating in an oscillatory state (regime 1).** 40% of GPe neurons (golden yellow) were converted into bursting neurons at time 1500ms—this had no effect of on the network activity state. To see the effect of spike bursting in STN neurons, in addition to the 40% GPe neurons, we also converted 30% of STN neurons (cyan) in bursting neurons at 3500ms. Even this change failed to alter the network activity state. The instantaneous firing rate (binsize = 10 ms) is plotted in black for bursting and non-bursty populations for GPe and STN.
(TIFF)

**S9 Fig. Effect of spike bursting when the network was operating in the transition regime (regime 2).** 40% of GPe neurons (golden yellow) were converted into spike bursting neurons at time 1500ms. This led to the emergence of weak beta band oscillations (see the spike raster in the right panel before 3500ms). To see the effect of bursting in STN neurons, in addition to the 40% GPe neurons, we also converted 30% of STN neurons (cyan) in bursting neurons at 3500ms. Spike bursting in STN quenched the oscillation initiated by spike bursting in the GPe. The instantaneous firing rate (binsize = 10 ms) is plotted in black for bursting and non-bursty populations for GPe and STN.
(TIFF)

**S10 Fig. Effect of spike bursting when the network was operating in an non-oscillatory state (regime 3).** 40% of GPe neurons (golden yellow) were converted into bursting neurons at time 1500ms—this had no effect on the network activity state. To see the effect of spike bursting in STN neurons, in addition to the 40% GPe neurons, we also converted 30% of STN neurons (cyan) in bursting neurons at 3500ms. Even this change failed to alter the network activity state. The instantaneous firing rate (binsize = 10 ms) is plotted in black for bursting and non-bursty populations for GPe and STN.
(TIFF)

**S11 Fig. STN spike bursting quenches oscillations by imposing a lower frequency on STN population.** At 3500 ms, an oscillation of 15Hz was imposed on STN population, instead of replacing STN neurons by bursting neurons. These change in the beta band oscillations because of the injection of 15 Hz oscillations in a fraction of STN neurons are qualitatively similar to the results show in Fig 4. These results explain how spike bursting in STN can quench oscillations when a small fraction of neurons are bursting.
(TIFF)

**S12 Fig. Spectrograms for single neurons with 40% of bursty neurons in GPe and 90% of bursty neurons in STN. (A)** Firing rate histograms of bursty (left) and non-bursty (right) neurons in the GPe. **(B)** Firing rate histogram of bursty (left) and non-bursty (right) neurons in the STN. For both GPe and STN we chose three exemplary neurons, #1—neuron with average firing rates $\leq$ mean population firing rate (37.16 spks/s), #2—neuron with average firing rate = mean population firing rate, #3—neuron with average firing rate > mean population firing rate. **C**: Spectrograms of each of the six chosen neurons from the STN and GPe.
(TIFF)

## Acknowledgments

We thank Drs. Arthur Leblois and Robert Schmidt for their input on the work.

## Author Contributions

**Conceptualization:** Jyotika Bahuguna, Ajith Sahasranamam, Arvind Kumar.

**Formal analysis:** Jyotika Bahuguna, Ajith Sahasranamam.

**Funding acquisition:** Arvind Kumar.

**Investigation:** Jyotika Bahuguna, Ajith Sahasranamam.

**Methodology:** Jyotika Bahuguna, Ajith Sahasranamam.

**Project administration:** Arvind Kumar.

**Resources:** Arvind Kumar.

**Supervision:** Arvind Kumar.

**Validation:** Jyotika Bahuguna, Ajith Sahasranamam.

**Visualization:** Jyotika Bahuguna, Ajith Sahasranamam.

**Writing – original draft:** Jyotika Bahuguna, Ajith Sahasranamam, Arvind Kumar.

**Writing – review & editing:** Jyotika Bahuguna, Ajith Sahasranamam, Arvind Kumar.

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
