## [Decision Letter · Decision Letter 0]

9 Sep 2019

Dear Dr Bahuguna,

Thank you very much for submitting your manuscript 'Uncoupling the roles of firing rates and spike bursts in shaping the GPe-STN beta band oscillations' for review by PLOS Computational Biology. Your manuscript has been fully evaluated by the PLOS Computational Biology editorial team and in this case also by independent peer reviewers. The reviewers appreciated the attention to an important problem, but raised some substantial concerns about the manuscript as it currently stands.

One major concern is what insight on actual biological mechanisms we can get from this work, which would make the difference in being published in PLOS Computational Biology.

A more extensive, and species specific, comparison, would be one of the points to address in this direction.

Still we appreciate that it could be unfeasible to address this task in the current framework, we leave to you the decision on whether you want to go for it.

We cannot, of course, promise publication at resubmission, and in case the actual biological insight is not clear (provided that all the other issues have been properly addressed), we could propose the transfer to PLOS One.

If you decide to resubmit, I would ask you to share the code with the editors and the reviewers already at that stage. This is not a formal requirement given that the policy of the journal is to have the code public in a third party repository at the moment of acceptance, but I hope you will agree that given that the code will be part of the paper, it makes sense to share it with the journal during the review process. In this phase the code can be in a private repository too.

Sincerely,

Daniele Marinazzo

Deputy Editor

PLOS Computational Biology

[LINK]

Reviewer's Responses to Questions

**Comments to the Authors:**

Reviewer #1: Firing rate vs. firing pattern is a perennial issue in neuroscience. The question of firing rate vs. firing pattern significance in Parkinson’s disease has been around for several decades. It was driven by clinical and experimental considerations: spiking units are recorded from patients or parkinsonian animal models, the next step one would do is to see if firing rate, bursting rate (or other measure of phasic and/or oscillatory activity) or both correlates with the symptoms. Over the years, the answer has shifted from firing rate to burstiness to oscillations and synchronization strength. However different studies point to different answers and the situation is somewhat messy. I think the prevailing consensus is that both firing rate and firing patterns are altered in Parkinson’s disease, but firing pattern (primarily oscillations and synchrony in the beta-band) are strongly related to some of the parkinsonian motor symptoms.

The issue is that while one may study experimental data for the best correlates of symptoms, firing rate and firing patterns are strongly connected. Uncoupling them in an artificial way in a model, does not clarify the mechanisms of pathological beta oscillations of Parkinson’s disease. Real GPe and STN neurons have multiple mechanisms, that connect spiking and bursting. For example, both cells express calcium currents and calcium-dependent potassium SK current. Therefore, the firing rate will affect calcium concentration, in turn affecting SK current, which will affect the bursting. The inverse is true as well, the bursting mechanisms will affect the number of spikes. As a result, the modelling framework of this study is poorly suited for investigation of the mechanisms of beta-band oscillations in basal ganglia. This problem (together with some of the modeling and data analysis problems enlisted below) make me think that this study should be rejected by PLOS Computational Biology. Perhaps it may go to some journal interested in computational models, not directly related to experimental neuroscience.

Modeling and data analysis:

Inter-spike interval in a burst in the model is 2ms (first paragraph of the materials and methods). This implies 500Hz firing rate within a burst, which is unrealistically high, especially for GPe. Moreover, there is a firing rate accommodation phenomena within bursts in GPe and STN, which will affect burst duration and frequency.

Why GPe and STN both have excitatory inputs and no inhibitory inputs? GPe receives massive inhibitory input from striatal MSN neurons.

STN affects GPe not only directly, but also via polysynaptic pathways (thalamocorticostriatal, thalamostriatal). Why these feedback circuits will not affect the results of this investigation?

What exactly does entropy Hs do? Does it capture periodicity outside of the beta band? What does it do for sharp and narrow vs broad and large peaks in spectra? Hs is a very important measure of neural activity in this study and its properties should be clarified.

The effect of non-monotonous dependence of the firing rates etc. – in the inhibitory-excitatory circuit like GPe-STN, these phenomena may strongly depend on the fine-tuning of synaptic coupling. Thus, some robustness study is merited.

The conclusion of GPe firing rates being not predictive of oscillations is very questionable. The model artificially disconnects firing rates and bursting/oscillations. In real neurons, they are not uncoupled and firing rate change may lead to (or result from) changes in bursting.

Moreover, contrary to what the manuscript seems to suggest, the experimental observations and modeling of “emergence of beta band oscillations is accompanied by a decrease of firing rate of GPe neurons” do not fit with the manuscript observations of GPe firing rate being not related to oscillations.

Parkinsonian changes resulting from dopaminergic degeneration may lead to oscillatory activity in different ways. Synaptic projections between GPe and STN are changing. But also cellular properties are changing resulting in firing pattern changes (for example, Luocif Woodhall Sehirli Neuropharmacology 2008). These dopamine-mediated effects are missing from the model. They can potentially be very important and alter the conclusions of the manuscript.

Presentation:

The presentation, especially in the abstract, gives a feeling of a very high confidence of the results as applied to the basal ganglia physiology in vivo. A fair manuscript would rather clarify that the results apply only to the model and whether the model is good or not is not really certain.

Beta band oscillations are indeed correlated with akinesia, but not with tremor. Ref 14 reports correlations of LFP in theta (not beta) band with tremor – first paragraph of introduction.

While several key modeling studies of oscillations in the basal ganglia are reviewed, other important work in this area is not: Humphries Stewart Gurney J Neurosci 2006, Park Worth Rubchinsky Physical Review E 2011, Merrison-Hort and Borisyuk Frontiers Comput Neurosci 2013.

Discussion of the beta oscillations origin misses a possibility of multiple mechanisms (Pavlides Hogan Bogacz PLOS Comput Biol 2015 and Ahn Zauber Worth Frontiers Comput Neurosci 2016).

Discussion of the birth of oscillations in inhibitory-excitatory networks and ref 32 -- oscillations in inhibitory-excitatory circuit is a half-center oscillator idea, which is probably a century old idea in neuroscience.

There seems to be some confusion about spike bursting vs. oscillations in this manuscript. It would help to clarify how precisely both terms are used here.

The idea that GPe-STN network may be operating in the regime at the border of oscillatory and non-oscillatory states seems to be very similar (although not identical) to an idea of GPe-STN network operating on the border between synchronized and nonsynchronzied dynamics (Park Worth Rubchinsky Physical Review E 2011). It would make sense to discuss how deep this similarity is.

Comparison of the burst duration length in mice and primates on page 10. Healthy mice case is reported, while ref. [13] reports human data in disease. Is it a fair comparison? Also, mice vs. primate data comparison is always a complicated issue.

Discussion of how the manuscript result can explain experiments with DBS does not appear to be convincing given how complex DBS is with stimulation engaging multiple fibers etc. (also, citation for STN firing rates there is missing).

Conclusion of the “Effects of changes in spike bursting…” section in the discussion: “we propose that increase in STN bursting might play a compensatory role…” The increase in bursting may be due to the lack of dopamine (see a ref to Luocif paper above and other experiments). Which appear to be very basic dopamine-mediated phenomena on a cellular level, not related to compensation of network changes.

Referencing to supplementary figures in the manuscript is broken.

Reviewer #2: The manuscript “Uncoupling the roles of firing rates and spike bursts in shaping the GPe-STN beta band oscillations” by Bahuguna and colleagues presents a comprehensive computational study trying to untangle the effects of two typical PD phenomena, the change in firing rate and bursting, on beta band oscillations. The study utilizes the State-dependent Stochastic Bursting Neuron (SSBN) which the group used in previous studies to separate the two components.

Major comments:

1. The effect is only in a small regime (osc/non-osc border). It is unclear whether this regime is relevant at all to the physiological state during PD.

2. The definition of beta is not consistent in the manuscript and varies across analyses, all the way from 10-35 to 15-30 and finally 15-20 (for example in beta bursts) while in the few figures that show the actual oscillations it seems like a narrow band around 20 Hz (or 15 Hz). Consistency and figures with less-process spectrograms are required.

3. The analysis of the spectrum utilizes the binned population, why weren’t individual neurons examined?

4. Results, Figure 3: It seems that the change in bursting changes the rate as well (which might be the actual one changing the oscillations). Isn’t this point the exact one that was supposed to be avoided?

5. It is unclear to what extent the bursting of the model SSBN resembles those of actual GPe and STN neurons.

6. Beta oscillations differ greatly between rodents and primates. However, in the current study the computational results are correlated with mixed species experimental results.

Minor comments:

1. Introduction, first paragraph: Death of the DA SNc neuron is only part of the story in PD (see Braak & Braak). The cognitive deficits and some behavioral symptoms are related to other processes associated with the disorder.

2. Introduction, first paragraph: Tremor is typically associated with non-beta oscillations.

3. Introduction, third paragraph: The definition of bursting and its separation from oscillations is not well defined in many of the rodent and primate studies mentioned.

4. The observed shifts in oscillation from 20 to 15 Hz are not observed experimentally. Moreover, perhaps the wide beta range in the study comprises of multiple smeared narrow bands?

Reviewer #3: Bahuguna and colleagues aim to dissociate the impact of firing rate and spiking patterns such as bursting on STN-GPE oscillations in the beta frequency band. These insights are quite important since beta oscillations play a central role in both physiological processes (e.g. motor control and impulsivity) and common neurological disorders (e.g. Parkinson’s disease).

Extensive work has been conducted on the oscillatory dynamics of the basal ganglia with a particular emphasis on the STN-GPe loop. However, this is one of the first works aiming to disentangle firing rate and bursting firing patterns. Authors also discuss implications of their work in the context of high conflict tasks (i.e. hold your horses) extending the scope of the paper beyond movement disorders.

The methodology is sufficiently explained, and the manuscript is well organized and clearly written. While I thoroughly enjoyed reading this paper, there are couple of points which would significantly improve the paper:

1) Inputs to the STN-GPe populations are uncorrelated Poisson processes and external inputs via the hyperdirect pathway and the indirect pathway are assumed to be tonic excitation and inhibition, respectively. In the context of Parkinson’s disease both neural populations are driven by rhythmic inputs and spiking activity are observed at certain phases of these oscillatory inputs (i.e. cortical beta). This issue is not only relevant for Parkinson’s disease but also for high conflict tasks. Several experimental studies have shown that during these tasks the mPFC exhibit rhythmic activity patterns (evoked in the theta band and ongoing oscillations in the beta band) and the STN spiking occurs at certain phases of these oscillations. While the ideal scenario would be for the authors to incorporate rhythmic inputs into their model, since this would be a significant undertaking and completely alter the paper - at the very least, the authors should discuss their work in the context of this literature and motivate the simplifications employed in their study.

2) Percentage of neurons that exhibit bursting spiking patterns is an important parameter in this study. The authors should discuss how neurons are recruited to exhibit these firing patterns (not how this switch is implemented but how it would take place in the context of the wider network).

Minor comment:

1) Reference missing – second paragraph of the discussion

**Have all data underlying the figures and results presented in the manuscript been provided?**

Reviewer #1: None

Reviewer #2: Yes

Reviewer #3: None

PLOS authors have the option to publish the peer review history of their article (what does this mean?). If published, this will include your full peer review and any attached files.

Reviewer #1: No

Reviewer #2: No

Reviewer #3: No

---

## [Decision Letter · Decision Letter 1]

24 Jan 2020

Dear Dr Bahuguna,

Thanks for this revised version, which successfully addressed most of the points raised. A couple of points remain outstanding, and you have the chance of doing so in a resubmission.

Sincerely,

Daniele Marinazzo

Deputy Editor

PLOS Computational Biology

Daniele Marinazzo

Deputy Editor

PLOS Computational Biology

[LINK]

Reviewer's Responses to Questions

**Comments to the Authors:**

Reviewer #1: The new version has been substantially revised. Many concerns have been extensively addressed. However, my major concern is that the study is trying to dissociate the phenomena that are inherently connected at the experimental level. In a modeling study one can isolate firing rate and firing pattern and explore their effects. I don’t see how relevant this would be to experimental context, where the firing rate and pattern are tightly connected on a very fundamental level. This is why, as I wrote earlier, this modeling approach may be an interesting exercise in computational modeling, but is remote from experiment. Therefore I don’t really see it as a PLOS CB paper. This is not a wrong study as a modeling exercise. But not more than that. I would expect to see it in a modeling journal or maybe PLOS One, which does not emphasize significance and experimental relevance.

Reviewer #2: The authors addressed most of the points raised regarding the original manuscript in a satisfactory manner. Two issues still need to be expanded:

1. The manuscript refers to multiple sources of physiological data mixing recording conditions such as in-vivo and in-vitro studies and species such as mice, rats, non-human primates and primates. There is a huge difference (much larger than the 10% variability tested for the model) between rodent species such as mice vs. rats and between different animal models of PD. It is crucial to separate the results and their implementation in the model and to state the data sources and differences clearly (results section) and discuss their implications on the model (discussion section).

2. The same issue still exists with beta band definition which is different in different species and (animal) models of PD. The authors should clearly state the relation to the specific beta band that they choose and their reasoning for it.

Reviewer #3: Authors have addressed all my concerns .

**Have all data underlying the figures and results presented in the manuscript been provided?**

Reviewer #1: None

Reviewer #2: None

Reviewer #3: Yes

PLOS authors have the option to publish the peer review history of their article (what does this mean?). If published, this will include your full peer review and any attached files.

Reviewer #1: No

Reviewer #2: No

Reviewer #3: No
---

## [Editor Report · Decision Letter 2]

25 Feb 2020

Dear Dr Bahuguna,

We are pleased to inform you that your manuscript 'Uncoupling the roles of firing rates and spike bursts in shaping the STN-GPe beta band oscillations' has been provisionally accepted for publication in PLOS Computational Biology.

Before your manuscript can be formally accepted you will need to complete some formatting changes, which you will receive in a follow up email. A member of our team will be in touch within two working days with a set of requests.

Best regards,

Daniele Marinazzo

Deputy Editor

PLOS Computational Biology

---

## [Editor Report · Acceptance letter]

13 Mar 2020

PCOMPBIOL-D-19-01260R2 

Uncoupling the roles of firing rates and spike bursts in shaping the STN-GPe beta band oscillations

Dear Dr Bahuguna,

I am pleased to inform you that your manuscript has been formally accepted for publication in PLOS Computational Biology. Your manuscript is now with our production department and you will be notified of the publication date in due course.

With kind regards,

Matt Lyles
